# Highly adaptable deep-learning platform for automated detection and analysis of vesicle exocytosis

Abed Alrahman Chouaib[1], Hsin-Fang Chang [1], Omnia M. Khamis [1], Nadia Alawar [1], Santiago Echeverry [2], Lucie Demeersseman[3], Sofia Elizarova[4], James A. Daniel [4], Qinghai Tian[5], Peter Lipp [5], Eugenio F. Fornasiero [6,7], Salvatore Valitutti [3], Sebastian Barg [2], Constantin Pape [8,9], Ali H. Shaib [6] ✉ & Ute Becherer [1] ✉

Activity recognition in live-cell imaging is labor-intensive and requires significant human effort. Existing automated analysis tools are largely limited in versatility. We present the Intelligent Vesicle Exocytosis Analysis (IVEA) platform, an ImageJ plugin for automated, reliable analysis of fluorescence-labeled vesicle fusion events and other burst-like activity. IVEA includes three specialized modules for detecting: (1) synaptic transmission in neurons, (2) single-vesicle exocytosis in any cell type, and (3) nano-sensor-detected exocytosis. Each module uses distinct techniques, including deep learning, allowing the detection of rare events often missed by humans at a speed estimated to be approximately 60 times faster than manual analysis. IVEA's versatility can be expanded by refining or training new models via an integrated interface. With its impressive speed and remarkable accuracy, IVEA represents a seminal advancement in exocytosis image analysis and other burst-like fluorescence fluctuations applicable to a wide range of microscope types and fluorescent dyes.

Live cell fluorescence microscopy plays a central role in the analysis of cellular dynamics including organelle and protein motion, as well as biosensor-based measurements of cellular activity, such as fluctuations in ion concentration or exocytosis events[1–3]. Upon expression of fluorescent sensors or incubation with specific probes, cellular activity can typically be observed as a sudden change in fluorescence intensity in recorded videos. While more sensitive and faster systems are being developed to acquire larger (terabytes) amounts of data at a faster

rate, fluorescence signal analysis is usually a challenging endeavor that requires extensive manual effort and becomes a bottleneck. An obvious challenge is to develop automatic detection methods for cellular activity that are sufficiently versatile and reliable enough to be easily implemented and applicable to batch analysis.

Regulated exocytosis, which is essential for cells to secrete substances, is a prime example of a rapid dynamic cellular event that is challenging to detect and measure. This difficulty arises from the

[1]Department of Cellular Neurophysiology, Center for Integrative Physiology and Molecular Medicine (CIPMM), Saarland University, 66421 Homburg, Germany. [2]Medical Cell Biology, Uppsala University, 75123 Uppsala, Sweden. [3]Cancer Research Center of Toulouse, INSERM U1037, 31037 Toulouse, France. [4]Department of Molecular Neurobiology, Max Planck Institute for Multidisciplinary Sciences, 37075 Göttingen, Germany. [5]Center for Molecular Signaling (PZMS), Institute for Molecular Cell Biology, Research Center for Molecular Imaging and Screening, Medical Faculty, Saarland University, 66421 Homburg, Germany. [6]Department of Neuro- and Sensory Physiology, University Medical Center Göttingen, 37073 Göttingen, Germany. [7]Department of Life Sciences, University of Trieste, 34127 Trieste, Italy. [8]Institute of Computer Science, Georg-August University Göttingen, 37077 Göttingen, Germany. [9]Cluster of Excellence 'Multiscale Bioimaging: from Molecular Machines to Networks of Excitable Cells' (MBExC), Georg-August-University Göttingen, 37077 Göttingen, Germany. ✉e-mail: ali.shaib@med.uni-goettingen.de; ute.becherer@uks.eu

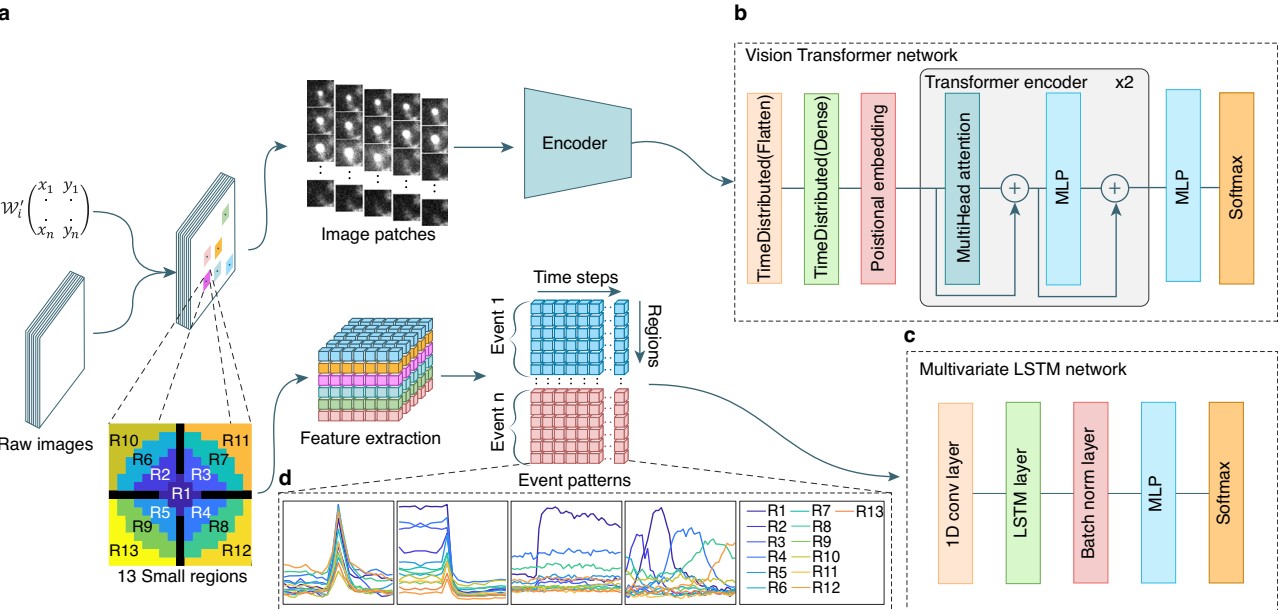

**Fig. 1 | Overview of our neural networks and the feature extraction process.** **a** Flow of the two distinct methodologies employed to prepare the data prior to classification. LM coordinate matrix $\mathcal{W}'_i \in \mathbb{N}^{2 \times d}$ represents the selected region. Methodology 1 (top) is used for random burst events. The regions are extracted and then fed into a shared encoder network. A total of 26 patches, each $32 \times 32$ pixel, were extracted for every selected region. Methodology 2 (bottom) is applied to stationary burst events, where feature extraction is performed. Feature vectors comprise 13 regions centered on the event's local maxima ($R_1, R_2 \ldots R_{13}$). The black regions in the "13 Small regions" scheme represent buffer zones. The feature extraction output for stationary burst events is organized by event count, region, and time series dimension. **b** Vision transformer network architecture. **c** Multivariate LSTM network architecture. **d** LSTM network recognizes each data package as a graph of 13 curves representing the regions normalized mean-intensity variations over time. The LSTM model is available for random burst event analysis as well. Event pattern graphs illustrate: 1. single T cell lytic granule fusion in which the fluorescent cargo was either pH-sensitive (left) or 2. pH-insensitive (middle left); 3. fusion at neuron synapse in which the vesicles are stained with pH-sensitive membrane protein (middle right); 4. fusion of a single moving granule (right).

considerable diversity of fluorescent signals that can be observed, for example, during synaptic transmission, release of hormones and cytokines, or targeted release of cytotoxic proteins by immune cells[4,5]. A commonly used approach to follow exocytic events occurring during neuronal synaptic transmission in live imaging relies on the expression of vesicular proteins tagged with pH sensitive fluorophores such as the super ecliptic pHluorin (SEP)[6–8] or on other types of markers such as the lipophilic FM dyes[9–11]. These can be imaged in different modalities including epifluorescence and confocal microscopy. To observe individual vesicle/granule exocytosis in endocrine or immune cells with high temporal and spatial resolution in real time, total internal reflection fluorescence (TIRF) microscopy[12–14] in combination with pH dependent and independent sensors is the method of choice. TIRF microscopy enabled the deciphering of trafficking and attachment of granules to the plasma membrane[9,15,16], and to follow precisely their fusion with the plasma membrane[17–20].

The development of rapid and reliable systems for the detection of exocytic events in the recorded videos should facilitate the development of therapies for mental disorders[21–23], diabetes[24] or immunological deficiencies such as hemophagocytic lymphohistiocytosis[25]. While several automated methods for exocytosis have been developed[26–32], their practical implementation has been limited either by their complexity or by their lack of versatility for a diverse range of datasets. One way to circumvent these limitations is the use of machine learning, particularly deep learning, which shines for its reliability and effectiveness. Deep learning is applied in different areas such as image segmentation, data analysis, or 3D model prediction[33–35]. Unlike mathematical models, which typically can only detect specific events, the appropriate use of deep learning can be leveraged to distinguish multiple types of events. In addition, deep learning can be used to

develop systems that adapt to a wide variety of data, enabling batch data analysis with minimal or no human input. Here we introduce an adaptive automatic vesicle fusion detection program named IVEA (Intelligent Vesicle Exocytosis Analysis; pronounced [ˈʌɪvi]), which uses deep learning to analyze a wide array of vesicle fusion events. IVEA is a trainable, modular tool that uses a hybrid approach based on the combination of computer vision and AI. The program is easily accessible as an ImageJ plugin (https://github.com/AbedChouaib/IVEA), and contains three completely independent detection/recognition modules that are optimized for analyzing the most common types of exocytosis events (Supplementary Fig. 1).

The first two techniques involve discriminator neural networks to detect the described burst events occurring during the exocytosis of fluorescently labeled vesicles. Module one is based on a vision transformer network (ViT)[36] (Fig. 1b) to visualize and classify exocytosis of individual vesicles/granules in any type of cells (Supplementary Fig. 1a–c). Module two is based on a long short-term memory (LSTM) network[37] due to its comparatively low computational resource demands (Fig. 1c). It is used to detect exocytosis that occurs during synaptic transmission in neurons (Supplementary Fig. 1d,e). Module three is designed to extract areas with fluorescent intensity variation using k-means clustering, and iterative thresholding (Supplementary Fig. 1f, g).

Due to its modular structure, IVEA has proven to be exceptionally versatile and reliable in the detection and classification of exocytosis events. It enables batch processing of data with a speed about 60 times faster than human experts. Finally, an additional user-friendly training platform allows users either to generate a new training model or to refine an existing pre-trained model, thus extending its application to other activity recognition tasks.

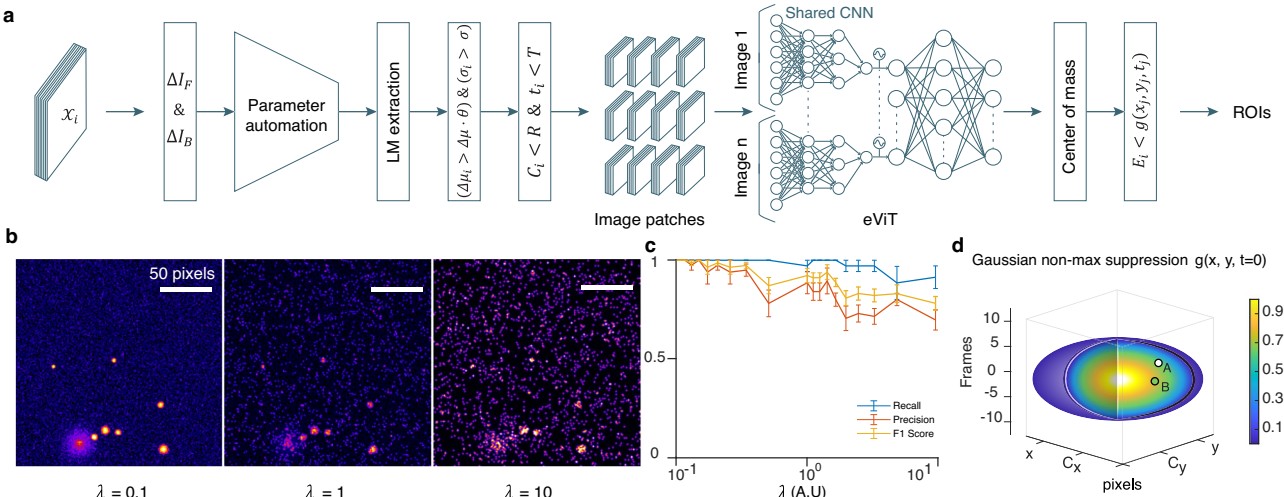

**Fig. 2 | Overview of random burst event detection algorithm and simulation analysis. a** Algorithm flowchart. $\mathcal{X}_i$ *denotes* the image sequence. $\Delta I_F$ and $\Delta I_B$ are the forward and backward subtraction images ($\Delta I$). $\Delta\mu_i$, $\sigma_i$, $C_i$ and $t_i$ are the event $E_i$ mean intensity, FWHM, center coordinate and the time for which $E_i$ was detected, respectively. $\Delta\mu$ and $\sigma$ are the automated parameters for detection. $\theta$, $R$ and $T$ *denote* the detection sensitivity, *a* search radius of *three* pixels, and *a* time interval of *four* frames, respectively. Image patches are the $32 \times 32$ pixels *crops* extracted over time for each selected region. The network scheme is an encoder-vision-transformer network (eViT). The center of mass step *refines* the *centroid of each* true positive event. The final step *applies a* Gaussian spatiotemporal function used *for* non-max*imum* suppression. **b** Effect of Poisson noise added to simulated videos. Left: simulated video with Poisson noise scaling factor $\lambda = 0.1$ with an ideal

exocytosis event in a cytotoxic T-lymphocyte. Middle: same video with $\lambda = 1$. Right: same video with $\lambda$ equal 10 times the signal. **c** Graph in the middle presents the evaluation of our eViT performance following the analysis of simulated videos with a noise scaling factor that varied between 0.1 and 10. The graph shows the average recall (blue), precision (orange), and F1 score values (yellow) with SEM ($n = 5$). **d** 3D ellipsoid representing the Gaussian spatiotemporal search equation $g(x_j, y_j, t_j)$ for event $E_j$. The color bar ranging from 0 to 1 corresponds to the mean gray value at point (x, y, t). Point A and B are two true positive events occurring close in time and space to $E_j$. At *point* A the mean gray value at $(x_A, y_A, t_A)$ exceeds $g(x_j, y_j, t_j)$, indicating that A is a separate event. While, at point B the mean gray value at $(x_B, y_B, t_B)$ is below $g(x_j, y_j, t_j)$, meaning $E_j$ = B. Source data are provided as a Source Data file.

## Results

### Classification of exocytic events for automated analysis

To develop an automatic analysis of exocytosis, it is necessary to first identify categories of events and then create software tools capable of recognizing them. In general, exocytic events can be classified into three main categories (Supplementary Fig. 1). The first two categories pertain to the measurement of vesicle exocytosis (Supplementary Fig. 1a–e), viewed as a transient change of fluorescence intensity on discrete spots. Therefore, we call these events, fluorescent burst events or burst events. When measuring the exocytosis of individual vesicles/granules, they often move towards the fusion site before exocytosis occurs (Supplementary Fig. 1a–c). This site can be anywhere on the plasma membrane, making multiple fusions at the same site unlikely. We termed this first type of event "random burst event". In contrast, synaptic transmission in neurons occurs exclusively at synapses and encompasses the fusion of multiple vesicles within a brief period, resulting in a change in fluorescence intensity at static positions. This is observed during spontaneous exocytosis as well as during repetitive stimulation (Supplementary Fig. 1d, e). Thus, the analysis can be conducted without considering moving fluorescent objects. This second type of events are referred to as "stationary burst events". Finally, a third category of events arise from the measurement of secreted transmitters using, for example, near-infrared fluorescent dopamine nanosensor 'paint' AndromeDA[38,39], and other composite nanofilms[40,41]. When the released neurotransmitter binds to the nanosensors around the synapse they emit a spreading fluorescent signal in a concentration-dependent manner (Supplementary Fig. 1f, g). We call these events "hotspot area" events. The detection of all these three different types of events requires different analytical approaches (Module 1 –3), involving the use of specific models (Supplementary Table 1). They are all included in IVEA, which runs in the widely used and open source Fiji program as a multi-purpose plugin with different applications in a comprehensive GUI (Supplementary Fig. 2). While IVEA is fully automated, users are offered the option to

fine-tune the plugin's advanced settings (Supplementary Fig. 3) or train the models to accommodate specific experimental paradigms.

### Event detection and classification paradigm

The IVEA software detects burst event activity in two phases. First, it automatically identifies intensity fluctuation and generates spatio-temporal coordinates (x, y, t) at the local maxima. Around each coordinates, IVEA defines a region of interest (ROI) and extracts a small image section (e.g., $32 \times 32$ pixels), along with frames captured before and after the identified peak (e.g., 10 frames on each side) (Fig. 1a). This yields a set of image patches that focuses on each ROI. In the second phase, IVEA classifies these image patches using neural networks that differ depending on whether the burst events are stationary or random. For random burst events (module 1), an encoder Vision Transformer (eViT) processes these image sequences to determine whether an exocytosis event has occurred (Fig. 1b). Due to the computational demands of the eViT when handling large numbers of image patches and extended sequences, we implemented an LSTM network for analyzing stationary burst events (module 2, Fig. 1c). Prior to their integration into the LSTM network, image patches undergo a reduction in dimensionality and a flattening process (Fig. 1a, d). This procedure enables the analysis of extended activity sequences while concurrently reducing the computational and memory overhead (see Methods section).

Detection and accurate prediction are two critical factors for analyzing fusion events. To evaluate our software, we compared the IVEA predictions with human experts' (HE) manual detection of different sets of videos acquired using different techniques. The videos employed in our results were devoid of events within the initial four frames of the acquisition (see Methods).

### Event simulation validation for random burst events (module 1)

The initial evaluation of the software's performance was conducted by the creation of simulated videos (Fig. 2b), employing a ground truth of

vesicles with an average radius of $2.7 \pm 0.36$ non-dimensional pixels. With this approach we emulated random burst events, as observed during the fusion of lytic granules in cytotoxic T lymphocytes (CTL) (see Methods). Subsequently, the aforementioned videos were analyzed with IVEA random burst event module (Fig. 2a) using average computers without relying on GPU (see Methods). We employed IVEA's default parameters with the exception of the neural network radius, which was set to 16 non-dimensional pixels. All simulated fusion occurrences were successfully identified without any false positive (FP) detection. To ascertain the limitations of IVEA accuracy, we have introduced white noise and Poisson noise over the videos (Fig. 2b) using Eq. (17) (see Method section).

The Poisson noise scaling factor λ was increased from 0.1 up to 10 times the signal. We ran IVEA on our simulated noise enriched videos and achieved recall of $99.71 \pm 0.29\%$, a precision of $94.49 \pm 3.23\%$ and F1 score of $96.71 \pm 1.91\%$ for $\lambda = 0.1$ up to $\lambda = 1$, which corresponds to a rather low SNR (Fig. 2c). When λ was increased further, IVEA began to lose some of the small vesicles, due to the added Poisson noise that surpassed the signal strength. For $\lambda = 1$ up to $\lambda = 10$, the recall was $96.86 \pm 2.55\%$, the precision was $79.22 \pm 4.68\%$ and F1 score was $86.51 \pm 3.27\%$. However, these are noise levels exceeding experimental TIRFmicroscopy videos, which would be analyzed by the scientist. Our results show that IVEA is ready to analyze real data, acquired with a challenging signal-to-noise ratio.

## Random burst events analysis (module 1)

We have analyzed several videos in which CTL secreted lytic granules (Fig. 3a, b and Supplementary Movie 1-4). In CTLs, lytic granules were stained via expression of granzyme B, a cargo protein, that was tagged with a fluorescent protein. This fluorescent protein was either the pH sensitive pHuji (Fig. 3a), the weakly sensitive eGFP or the pH insensitive tdTomato (Fig. 3b) generating very distinct event signatures as displayed in Supplementary Fig. 1b, c. The videos were recorded at 10 Hz over a duration of 10 min and encompassed 1 to 33 CTLs. In these videos the total number of fusion events detected by the HE was around 245 (Fig. 3a, b, Supplementary Movie 1–4). The time that the HE required to analyze one cell was 10 min for lytic granules labeled with pH-insensitive fluorescent protein and 5 min for lytic granules labeled with a pH-sensitive fluorescent protein. This amounts to 300 min of analysis for videos containing 30 cells. IVEA required less than a minute per cell with a video size of ~256 × 256 pixels and 3000 frames. On a computer equipped with an Intel Core i9 10th generation, this sums up to about 15 min for the entire video with 30 cells irrespective of the fluorescent marker protein. The performance of IVEA platform was further evaluated on CTL datasets acquired with different marker proteins (Fig. 3c) or using the small fluorescent label LysoTracker Red (Supplementary Fig. 4) that display very different fusion kinetics. The latter dataset was acquired in a separate laboratory equipped with a different TIRF microscope. To increase signal diversity, we also analyzed random burst events in chromaffin cells (Fig. 3d) and INS-1 cells (Fig. 3e) that secrete dense core granules. In these cells the vesicles are smaller than in CTLs and they are often more packed. The granules were stained via the over-expression of NPY attached to a fluorescent marker that are weakly pH sensitive like mGFP or mNeonGreen or pH insensitive such as mCherry.

The eViT network for random burst events was trained with 10 distinct categories. The event categories encompass three distinct types of exocytosis events: fusion with a cloud of spreading fluorophore, fusion without a cloud (sudden disappearance), and latent granule fusion (abrupt fusion onset); and 8 other types of events, such as fast drift or focus change, granule movement, random noise, random noise with intensity fluctuation, granules with noise, and granule docking and undocking. Analyzing the videos by HE identified 770 fusion events in all videos, across all cells and labels. IVEA detection routine registered around 156k ROIs for later

classification, out of which 2418 were identified as true events by our eViT network. If a random burst event exhibits a spatial spreading of fluorescence (Supplementary Fig. 1b, c, Supplementary Movie 1–4), then a single event can be detected multiple times (duplicates). To address this problem, we have devised a new method that implements 3D Gaussian spread over time to eliminate the redundant detections (Fig. 2d, Eq. 10). After applying our non-max suppression algorithm (Eqs. 10, 11), 1025 TP events were selected while 1393 duplicates were discarded. Therefore, IVEA detected 255 additional events from those originally identified by the HE. All TP events were again manually validated by the HE to evaluate the network performance (Supplementary Table 2). The events originally missed by the HE were either small and weak events difficult to visually detect or simply overlooked, possibly due to the HE's limited attention span.

We evaluated two neural network architectures, eViT and LSTM, for detecting exocytosis in random burst events analysis (Fig. 1a, Supplementary Table 1). All TP events identified by the eViT, and the LSTM were verified by the HE. The analysis was conducted on the described videos. The results were divided according to the cell type and granule label. The model utilized was the GranuVision2, except for the CD63-pHuji, for which the GranuVision3 was used (Fig. 3). The GranuVision2 model was trained to differentiate between fusion with and without a cloud, while the GranuVision3 was trained on both phenomena, incorporating the latent granule fusion. The vision radius surrounding an event was adjusted according to the granule and its fusion size (see Supplementary Fig. 5). For instance, a radius between 14 and 16 pixels was used with videos that had a pixel size of 110 nm (Fig. 3a–c), while a radius between 7 and 12 nm was used for videos with a pixel size of 130 or 160 nm (Fig. 3d, e). A summary of recall, precision, F1 scores, and the number of events detected by eViT, LSTM, and human experts (HE) is provided Supplementary Table 3 (see Supplementary Table 4 for statistics).

The first set represents CTL with pH-sensitive staining (Fig. 3a). The results yielded a total of 85 TP events identified by the eViT and 70 by the LSTM, in comparison to 77 events identified by HE. The eViT achieved the best F1 score of $97.81 \pm 0.98\%$ (Supplementary Table 3, 4).

For set two, which represent CTLs with pH-insensitive staining (Fig. 3b), the eViT detected 219 TP events, while the LSTM identified 172, in comparison to 168 events found by a HE. The eViT achieved an F1 score of $89.31 \pm 2.12\%$ (Supplementary Table 3, 4). The eViT also detected proficiently the exocytosis of lytic granules stained with LysoTracker Red (Supplementary Fig. 4).

The third set is for latent granule fusion (Fig. 3c), where the CTL were stained with pH-sensitive CD63-pHuji or CD9-SEP. The eViT detected 96 TP events, the LSTM detected 78 compared with 86 events identified by a HE. The eViT demonstrated an F1 score of $98.16 \pm 1.30\%$.

The fourth set comprising chromaffin cells and INS-1 cells for smaller granules with challenging exocytosis features, stained with pH-insensitive probes (Fig. 3d, Supplementary Fig. 6). The eViT performed significantly better than the LSTM (Supplementary Table 3, 4). The eViT detected 412 TP events, the LSTM detected 110 compared with 292 events identified by a HE. The eViT demonstrated by far the highest F1 score of $86.58 \pm 9.81\%$.

For the last set, which represents INS-1 cells stained using NPY-mNeonGreen or NPY-mGFP (Fig. 3e) the eViT identified 214 TP events, in comparison to 66 by the LSTM, while the HE detected 147 events. The eViT recorded an excellent F1 score of $93.18 \pm 2.06\%$ which is more than twice as good as the F1 score of the LSTM (Fig. 3e, Supplementary Table 3, 4).

Overall, the average results across all sets show that the eViT outperforms the LSTM network, as the eViT achieved an average recall of $98.95 \pm 0.40\%$, with average precision of $88.94 \pm 3.64\%$ and average F1 score of $93.13 \pm 2.05\%$. In contrast, the LSTM had an average recall of $64.87 \pm 13.18\%$, an average precision of $86.87 \pm 3.05\%$ and an average F1 score of $68.58 \pm 10.16\%$. Therefore, we opt to choose the eViT model

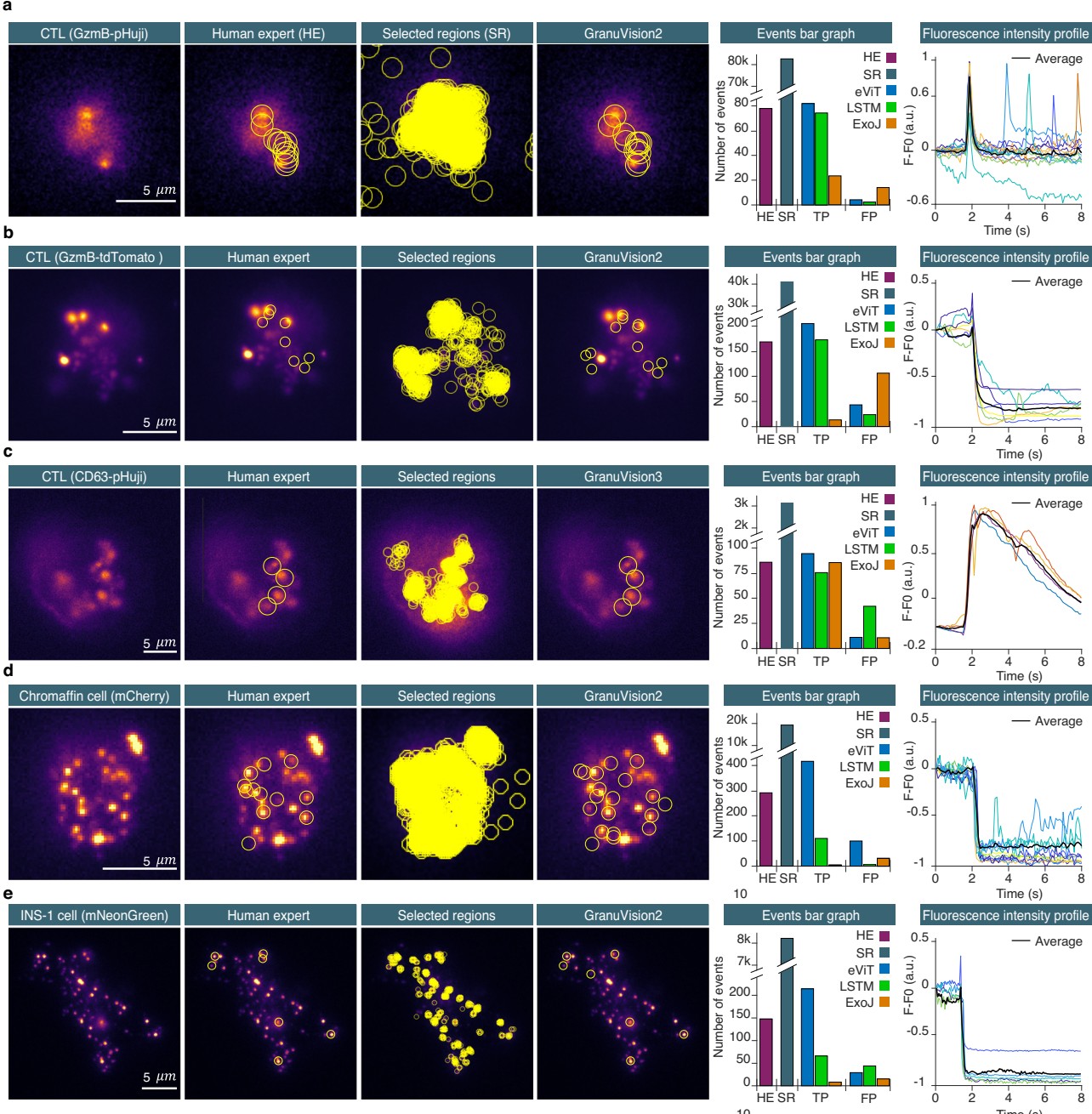

**Fig. 3 | Random burst event analysis.** Detection process compared to the human expert (HE) displayed from left to right; ROIs are shown as an overlay. Panel (**a–e**) from left to right, the first column shows TIRF-microscopy raw images of cytotoxic T cells, chromaffin cells, and INS-1 cells. The second column shows ROIs of the detected events by HE. The third column shows the raw images with all the selected regions prior to classification. The fourth column shows the classified events by our eViT network using the model indicated above. Bar graphs displays: total number of events identified by HE (purple), selected regions (SR, gray), events classified by eViT (blue), LSTM (green), and ExoJ (orange). The line plot column shows fluorescence intensity profiles of the true positive events detected by IVEA. The events profiles are aligned to their respective detection times. The fluorescence peaks at 2 sec corresponds to the exocytosis of the detected events. **a** CTL expressing pH-sensitive granzyme B-pHuji with thirteen movies of individual cells were analyzed ($n_{cell} = 13$). **b** CTL expressing pH-insensitive granzyme B-tdTomato. Seven videos, each containing 1–11 cells, were analyzed ($n_{cell} = 33$). **c** Shown are CTLs transfected with CD63-pHuji. The analysis was performed on five of the same type of movie and one movie of HeLa cell expressing CD9-SEP found in *Zenodo*[30]. **d** Chromaffin cells expressing NPY-mCherry (pH-insensitive fluorescent protein). Five videos were analyzed and pooled with five videos of INS-1 cells expressing NPY-mCherry ($n_{cell} = 10$). **e** INS-1 cells expressing NPY-mNeonGreen. Nine videos of individual INS-1 cells expressing NPY-mGFP or NPY-mNeonGreen were pooled (both weakly pH-sensitive yet displays distinct cloud release) ($n_{cell} = 9$). INS-1 cell videos were acquired at the Medical Cell Biology, Uppsala University, Sweden. The exocytosis-stimulation protocol is provided in the Methods section (acquisition protocol). Source data are provided as a Source Data file.

as the main model for the IVEA software for the random burst events analysis.

In addition, we down-sampled 10 Hz videos to 1 Hz to test IVEA performance at low image acquisition frequency. IVEA remains capable of detecting exocytosis at lower frequencies, but as the acquisition rate decreases, fast fusion events are naturally eliminated due to reduced temporal resolution, making them undetectable both manually and computationally (Supplementary Fig. 7a–c).

IVEA was compared to an existing ImageJ open-source plugin called ExoJ[42]. ExoJ was originally developed and validated on data that included CD63-pHuji and CD9-SEP labeling, among other stains, as published by Liu et al.[42]. When we tested ExoJ on our CD63-pHuji dataset and dataset from Liu et al., ExoJ achieved an F1 score of 88.92 ± 7.28%, which was comparable to the score of GranuVision3, albeit slightly lower (Fig. 3c). However, when tested on probes that have little or no sensitivity to pH (Fig. 3b, d, and e), ExoJ did not effectively detect exocytosis events, demonstrating its limited applicability beyond pH-sensitive labeled burst events (Fig. 3b, d, e). Even for GzmB-pHuji, a pH-sensitive probe (Fig. 3a), ExoJ struggled due to differences in intensity profiles. We tested it using default parameters, followed by a subsequent optimization process, but the results didn't improve much (Supplementary Tables 3–5). This indicates that ExoJ's rule-based approach is constrained by predefined detection criteria. Additionally, we tested pHusion, another program based on mathematical model[43]. This program did not yield better results than ExoJ (see Supplementary Table 6).

ExoJ demonstrated an excellent performance comparable to that of IVEA on a specific type of exocytosis (latent granule fusion), while failing to detect other types. In contrast, IVEA's deep learning-based approach, powered by eViT, successfully demonstrated a capacity to detect exocytosis events across both pH-sensitive and pH-insensitive datasets. By learning complex spatiotemporal patterns via new training or by adapting pre-trained models via a refinement process, eViT adapts to diverse fluorescence signals (Supplementary Fig. 8). Through the refinement of the model GranuVision3 on only 30 calcium sparks events measured with Fluo4 in cardiomyocytes[44], we were able to detect 298 TP and 2 FP events in comparison to 200 prior to refinement. This makes IVEA an extremely versatile and robust detection framework suited for a broader range of experimental conditions.

IVEA's output consists of two files sets: an ImageJ ROI zip file and two analysis CSV files. Each ROI is labeled and positioned on the center of mass at the peak of the fluorescence intensity of the event to which it corresponds. The CSV file contains measurements of the fluorescent intensity of each event over fixed time intervals, with their $x$ and $y$ center of mass coordinates, their full width at half maximum (FWHM), and the fluorescence intensity kinetics (rise time, decay and temporal FWHM).

## Stationary burst events analysis (module 2)
Stationary burst events were analyzed using the second branch of our platform (Fig. 4a), which employs the LSTM neural network for classification (Fig. 1a, c, Supplementary Table 1). The rationale behind this choice was conservation of memory and computational power. Our eViT for random burst events requires an input sequence of 26 image patches with a size of 32 × 32 pixels. This necessitates high memory and computational power both during the training phase and classification task. In the case of stationary burst events, the number of frames per sequence required to analyze an exocytosis event would need to be significantly higher than for random burst events due to different signal kinetics. A reasonable number of frames to study them would be between 41–100 frames. Furthermore, the number of selected regions to extract the image patches is high, with up to 14,000 ROIs in a single video. This would result in huge memory demands, necessitating the use of high-end computational resources to analyze stationary burst events with our current eViT.

We analyzed DRG neurons videos expressing SypHy that were derived from Shaib et al.[45] and Staudt et al.[46]. These videos display a variety of synapse count, intensity, vesicle movement, and background activity. In our data set, stationary burst events were characterized by fast rise (within 4.1 s, i.e. 41 frames acquired at 10 Hz, Supplementary Fig. 9d) of the fluorescence intensity in a spot like area (Supplementary Fig. 10a). Our neural network input layer for the stationary burst events was adapted for the input vector $\mathbf{P}(\mathbf{x}(t), \mathfrak{f}) \in \mathbb{R}^{T \times \mathfrak{f}}$, where $\mathbf{x}(t) \in \mathbb{R}^T$ as $\mathfrak{f}$

is the number of regions and $t$ is the time series for $0 < t \leq T = 41$ with $t \in \mathbb{N}$. Thus, the LSTM network discards the majority of the selected events stored whithin matrix $\mathscr{W}'$ (i.e. the selected regions, Fig. 1a) resulting in the classification of highly probable true events. For events in which the rise time was longer than 41 frames, the LSTM network sorted the events into the "intensity rise" category (Supplementary Fig. 9c) thereby discarding them from the collection of true events. For experimental conditions in which long lasting events with slow kinetics are the result of long stimulation paradigms or very high acquisition frequency (Supplementary Fig. 10), we adapted an "add frame" option that allows the user to adjust the event's time window by increasing the number of frames by $t_n$ (see method section). The videos that were analyzed, were acquired at 10 Hz and comprised 3000 frames, each measuring either 512 by 512 pixels[45] or 512 by 256 pixel[46]. The DRG neurons were stimulated electrically for 1 min[45] or they were stimulated twice for 30 s and 1 min with 10 s recovery phase in between[46]. The human analysis of these videos was a challenging task that required an average of 60 min per video. Detected events had predominantly a high fluorescence intensity variation or lasted for relatively long periods of time. IVEA reduces the time of analysis of the same videos to under 1 min per video. Furthermore, batch analysis capabilities exclude the need for manual parameter adjustment, as the tool automatically adapt to the input video's characteristics.

The results of the analyzed videos show that the neural network was able to classify virtually all the human labeled regions (Fig. 4b, c). Additionally, the neural network was able to detect more true fusion events than HE had originally detected by double checking and validating them as real events. The HE was able to detect overall 356 fusion events, while IVEA detection routine registered 84k ROIs for later classification. Most of these events were identified by the neural network as false events. A total of 2049 events were classified as true events, while only 70 events were identified as FP (Fig. 4c). The average recall, precision and F1 score were 88.12 ± 2.70%, 96.37 ± 0.45%, and 91.83 ± 1.61% respectively. Importantly, in comparison to the HE approaches, the LSTM network for stationary burst events could detect weak events or events with fast kinetics (Supplementary Fig. 9e, f; Supplementary Movie 5). Conversely, the HE was able to detect events with very slow kinetics (Fig. 4b) (Supplementary Fig. 10b). Due to the "intensity rise" category (Supplementary Fig. 9c) these slow events (i.e. longer than 41 frames) were missed by IVEA in the videos from Staudt et al.[46], in which long stimulation paradigms were applied to the cells, yielding more long-lasting events. However, they were detected when applying the "add frame" option that extends the time interval by adding 60 frames for correct classification (Supplementary Fig. 10, Supplementary Movie 5). Overall, IVEA identified about 5.4 times as many true events as HE. We compared IVEA to the existing open source software SynActJ[32] that was devised to analyze synaptic activity in neurons stained with the overexpression of synaptobrevin-SEP or alike proteins. First, we compared the result of IVEA and SynActJ on the provided test movie. SynActJ and IVEA were able to identify most of the active synapses but missed one and yielded one FP event (Supplementary Fig. 11a). Then, we compared both programs on one of our movies in which synaptic activity was clearly detectable by HE. While IVEA detected all events without any FP events, SynActJ was able to detect only a very limited number of active synapses and showed a significant number of FP events (Supplementary Fig. 11b). Thus, IVEA is by far superior to SynActJ.

For advanced analysis, IVEA distinguishes between various event types and categorize them based on their timing in respect of the experimental stimulation paradigms. The events can be classified as synchronized to the stimulus or unsynchronized (Fig. 4c). This feature was implemented as both types of events might show differences in kinetics. Stimulation time can be set manually, but to ease usability we also implemented an automatic stimulation detection. Our neural network was trained on $n_c = 9$ distinct events categories. These

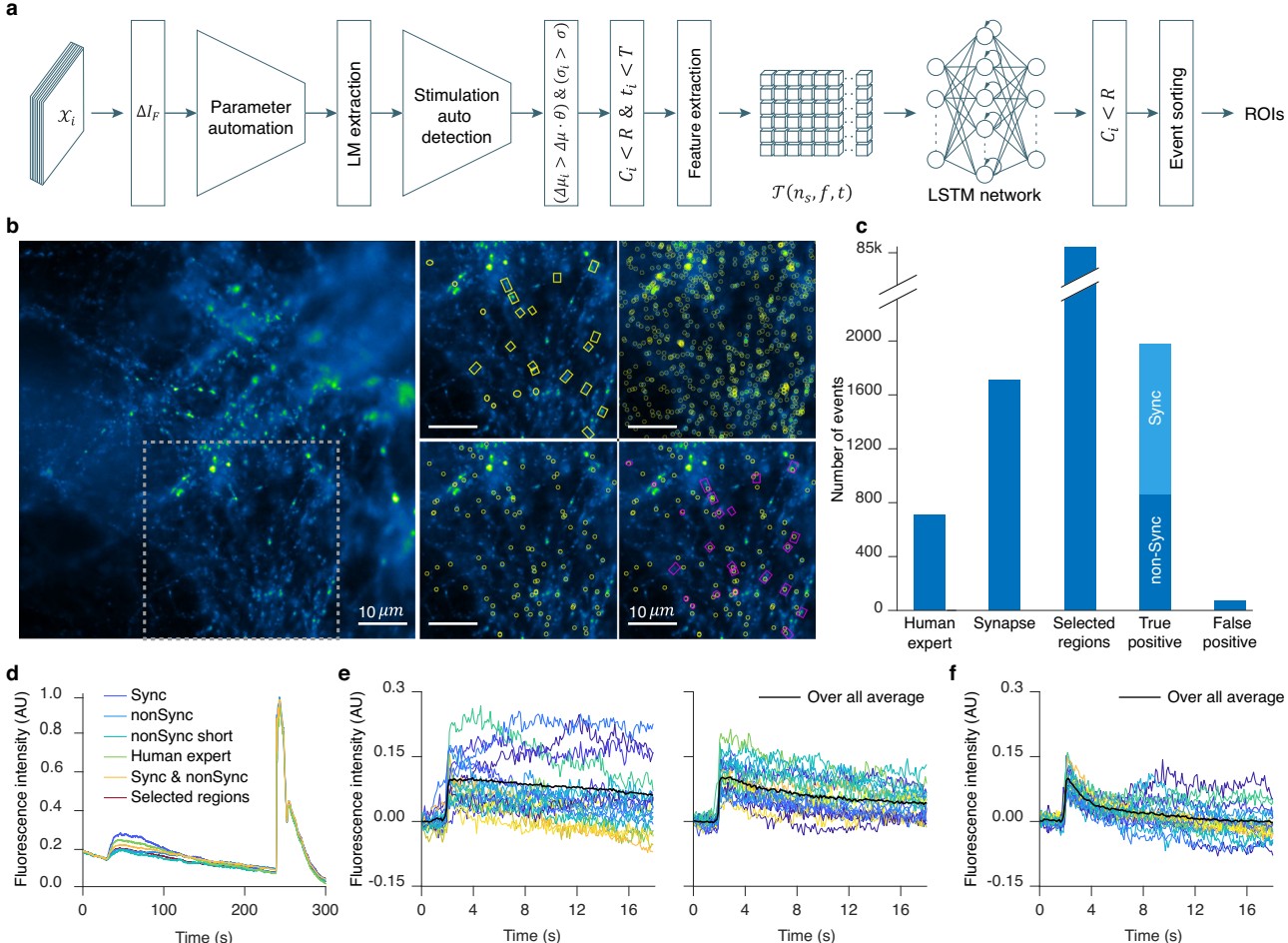

**Fig. 4 | Detection and analysis of exocytosis in neurons with stationary burst event algorithm. a** This panel displays the stationary burst event algorithm flowchart. Defining parameters: $\mathcal{X}_i$ denotes the raw images; $\Delta I_F$ is the forward-subtracted image. $\Delta \mu_i$, $\sigma_i$, $C_i$ and $t_i$ are the event $E_i$ mean gray value, full width at half maximum (FWHM), center coordinate, and event occurrence time; $\Delta \mu$ and $\sigma$ are the mean gray values and FWHM thresholds. $\theta$, $R$ and $T$ denote the detection sensitivity, search radius and the event time-interval, respectively. $\mathcal{T}(n_s, \mathfrak{f}, t)$ represents the extracted data in 3D tensor form. **b** Left, raw image of dorsal root ganglion (DRG) neurons over-expressing SypHy forming synapses on spinal cord neurons. Exocytosis stimulation protocol is given in the Methods under acquisition protocol. Right, depicts the area within the dashed gray box on the raw image overlaid with four different ROIs. Displayed, from left to right, top to bottom, are the human expert (HE) ROIs, the selected regions (SR) ROIs, the neural network ROIs, and a composite overlay of HE (Magenta) with neural network (Yellow) ROIs. **c** Bar graph representing the total number of events analyzed in 11 DRG neurons videos. IVEA parameters for the analysis were set to default. **d** Overall mean intensity profile of the combined ROI areas, comparing different event types shown in different colors as indicated. **e** Mean intensity profile over time representing the events detected at the stimulation time (synchronized events, left), and before or after stimulation (non-synchronized, right). **f** Mean intensity profile for short event category, whether synchronized or non-synchronized. The event intensity profiles are aligned on their respective detection time (**e & f**). Colored lines represent different events, while the thick black line shows their average. Source data are provided as a Source Data file.

categories include: four types of events (fusion, short time fusion ~ 4 frames (0.4 sec at 10 Hz), electrical or agonist stimulation, and $NH_4^+$ treatment) and five types of artifacts (fluorescence intensity rising, out-of-focus artifacts, vesicle motion, white noise, and fluorescence intensity fluctuations).

While events are sorted and labeled, spatial recognition task is performed to locate and unify the event's spatial identity while counting how many times the same synapse was active. Finally, the output for the stationary burst events are ROIs files whereby each ROIs is positioned at the event's maximum intensity temporal occurrence. The ROIs are labeled with an ID, frame number and event status. Additional outputs are the summary for each event and the measurements, such as intensity over a specified time interval and over the full video length.

### Hotspot area extraction (module 3)

For the third analysis conducted using IVEA, we employed distinct techniques from those mentioned earlier. In this analysis, we assumed

that the sensor array was in a fixed position, awaiting the occurrence of hotspots. Due to the limited availability of training data and the simpler features compared to the stationary and random burst events, we opted not to implement a neural network. Instead, we utilized k-means clustering and iterative thresholding for detection, and employed spatial search with mean intensity tracking over time for event recognition (Fig. 5a). Before applying the foreground detection method, we perform intensity fluctuation correction. The challenge while correcting the intensity fluctuation is to avoid altering and distorting the event as much as possible. To address this, we have developed a new method utilizing k-means combined with simple ratio for pixels value grouping and adjustment (see Methods). We call it "Multilayer Intensity Correction" (MIC) (Supplementary Fig. 12). The main idea of our method is to perform pixel value correction based on the variation of the average value of each cluster of pixels. To preserve the signal intensity, the signal should be a small range of pixels registered in a cluster, otherwise the signal is affected by the average value adjustment. MIC algorithm is performed through segmenting a given

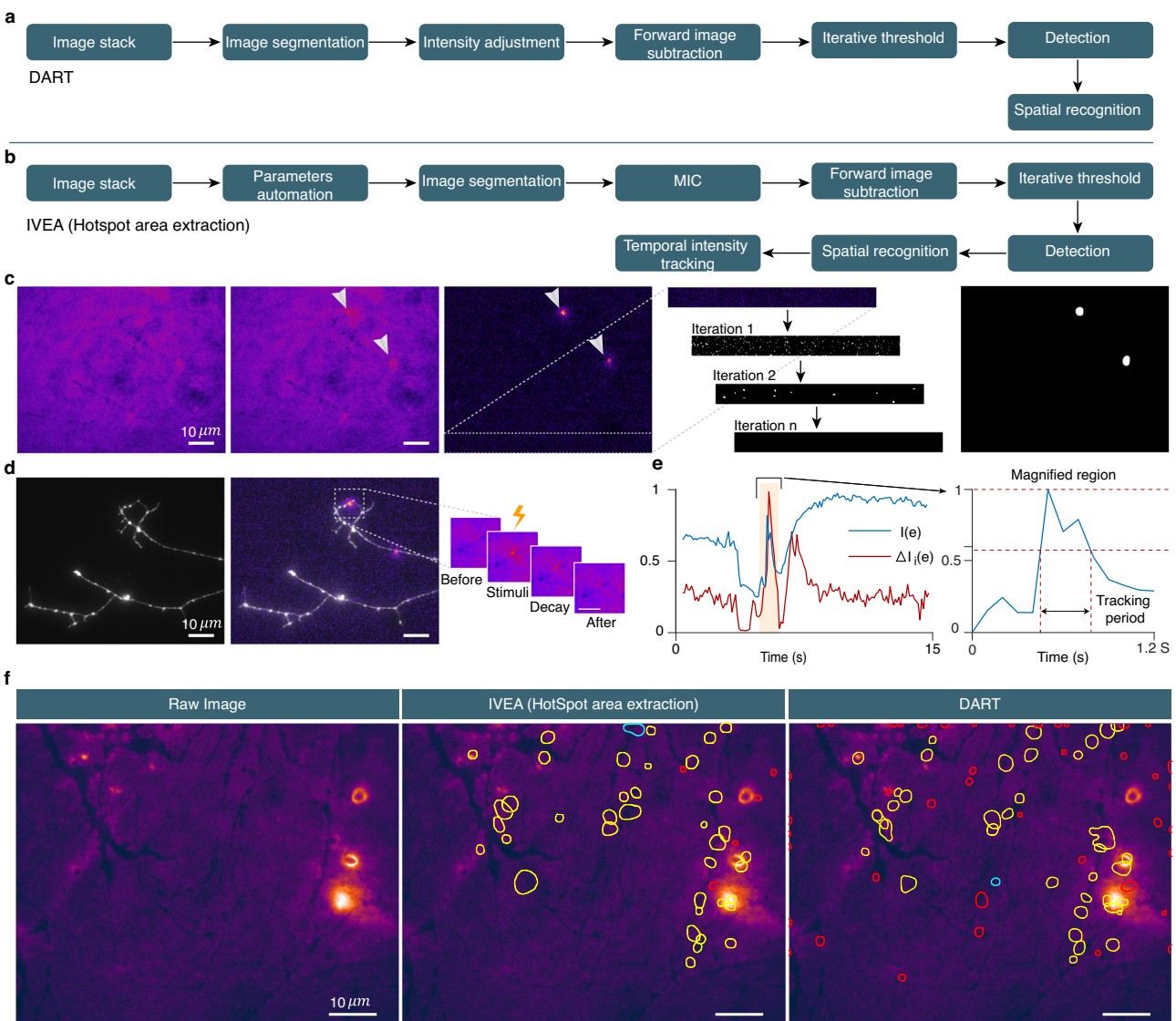

**Fig. 5 | Overview of the sensor-based exocytosis detection algorithm.**
**a** Algorithm flowchart of DART. **b** Algorithm flowchart of IVEA hotspots area extraction. IVEA employs three new methods compared with DART. These include parameter automation, multilayer intensity correction (MIC) (see Supplementary Fig. 12), and temporal tracking. **c** Detection process displays (left to right): raw image; hotspot mask (two hotspots); intensity variation image where hotspots are evident; iterative threshold steps over a cropped region of the intensity variation image; and the segmented image used to determine the events ROIs. **d** Event activity (left to right): raw image of dopaminergic neuron; intensity variation image into *k* layers using the k-mean clustering algorithm[47] after per-

composite with the raw image; sequence of zoomed snapshots that display the hotspot over time. **e** Graph representing the intensity variation over time for ROI's mean intensity of the original image sequence ($I(e)$, blue line) and for the intensity variation processed image sequence ($\Delta I_i(e)$, red line). The right-hand graph magnifies $I(e)$ around the hotspot occurrence window. It displays the temporal intensity tracking period. **f** Images comparing IVEA hotspot area extraction with DART. The yellow ROIs denotes the hotspot; red ROIs indicates probably false hotspots; and the cyan ROIs are true hotspots detected by one algorithm but not by the other. This figure displays images from two representative videos out of 8.

forming Gaussian filter of sigma equal to 1. After performing foreground detection, we extracted the hotspots from the processed image $\Delta I$ by converting it to a binary image using a global threshold (Fig. 5b). Since $\Delta I$ is the resulting matrix of intensity variation between two images, applying different types of threshold algorithms specially those found in Fiji, lead to unpredictable results. Therefore, we implemented our own iterative global threshold (Fig. 5c). The iterative threshold is not dependent on the statistical information calculated from the image. Instead, it is iterated over the noise level $\Delta I$, with the objective of eliminating it (Fig. 5b). This allows us to determine the global threshold for the fluorescence intensity value (see Methods). After detection of the possible hotspot, the fluorescence intensity of each event is temporally tracked (Fig. 5d). When the fluorescence intensity of the event falls below the mid intensity, the event signal is

considered to have disappeared and the tracking stops (Fig. 5e). The IVEA software, which uses advanced algorithms and automated parameters, reduced the need for users to iteratively adjust the parameters for analysis. Furthermore, it enhances precision compared to the previously used DART software with default parameters (Fig. 5f).

## Discussion
IVEA is a robust, pre-trained plugin for activity recognition, dedicated to the detection and analysis of vesicle exocytosis. It seamlessly integrates into the Fiji platform for open-source accessibility. IVEA achieves an impressive performance with F1 scores as high as 98 ± 1% for the eViT (random burst events, Fig. 4) and 94 ± 1% for the LSTM neural network (stationary burst events, Fig. 5). Furthermore, IVEA is proficient in distinguishing real events from artifacts such as photon shot noise or fast transient focus change (Supplementary Fig. 9a).

Three different algorithms can be selected by the user, making IVEA highly compatible with a range of exocytotic events displaying varying fluorescence intensity profiles (Supplementary Table 1). Additionally, IVEA's high adaptability results from its machine learning foundation, particularly, leveraging deep learning ViT[36], CNN and sequential models[48]. This enables IVEA to not only identify a wide range of exocytosis events, but also to learn and recognize new patterns, thereby expanding the scope of its capabilities.

Other Fiji-based exocytosis analysis plugins such as PTrackII[31] SynAct[32] or ExoJ[30] have been developed. Their algorithms rely on fixed mathematical and/or morphological models that are proficient at detecting a limited range of events as these algorithms cannot adapt and learn by themselves. The detection parameters can be adapted to a certain degree by adjusting complex input parameters that can be overwhelming for users with limited programming skills. This also precludes easy implementation of batch analysis when the analyzed videos have different characteristics, such as varying noise levels or fusion kinetics. Other programs like pHusion, have been developed but are only available in MATLAB, Python, or other proprietary platforms, making them less accessible[27–29,43]. Finally, an algorithm has been developed with the variability of the signal in mind, using convolutional neural network (CNN)[26], which is not made available as a software. In contrast, IVEA – a "plug and play" plugin, distinguishes itself by the adaptability of the detection and classification capabilities that is based on multivariate LSTM[49] and ViT[36] models. For stationary burst events, we chose LSTM network for the analysis, as it requires less memory usage when compared to eViT. This is particularly important as the number of extracted sequence patches and the number of frames analyzed per sequence is high. An analysis of all these sequences by eViT would require huge computational resources that cannot be provided by CPU computation and a reasonable RAM size (about 64 GB). In contrast, for random burst events, the presence of motion and other variables makes the classification of the events complex. This necessitates the use of more sophisticated models, such as the ViT network. To support this model, an encoder network was adapted and implemented to extract features prior to the ViT network. The added layers to the ViT model[36] was demonstrated using an ablation study (Supplementary Fig. 15) (see Method).

To assess IVEA's versatility, we extensively trained and evaluated it on exocytotic events in CTL, chromaffin cells, INS-1 cell and DRG neurons. Consequently, we were able to detect exocytotic events with high effectiveness, achieving a recall of 98 ± 4% and F1 score of 90 ± 6%, even in videos with low SNR and events presenting minimal features (dense core granules in INS-1 cells labeled with the pH-insensitive NPY-mCherry). Additional challenges such as granule clustering do not impair IVEA's capability to identify weak exocytotic events (Supplementary Fig. 6). Finally, IVEA successfully detected exocytotic events with different signatures than those used for training, albeit with some additional FP (Supplementary Fig. 4). However, IVEA is not designed to detect non-burst events such as those recorded in hippocampal neurons labeled with Synaptobrevin-SEP, FM1-43 or CypHer5E. In the case of Synaptobrevin-SEP stimulated exocytosis at synapses results in long-lasting increased fluorescence spreading out of the synapse[50,51], while with FM1-43[52] and CypHer5E[9], it produces only a very slow decay of fluorescence. In the future, adapting the eViT capability and retraining the new neural network may enable this type of analysis. However, this will require advanced computational capabilities and different dataset labeling. In contrast, we foresee that IVEA would have minimal difficulty of detecting synaptic transmission measured via iGluSnFR3[53]. While IVEA is virtually universal for detecting burst exocytosis in a wide array of experimental paradigms with the current trained models, users might still encounter specific needs. Therefore, we provide Python scripts with a simple GUI for training purposes, along with a configuration JSON file for parameter adjustments. The choice to use Python was motivated by challenges related to

vectorized computations, alongside variations in the versions of C++ jar files across Fiji, Google TensorFlow, and the deeplearning4J library. These scripts facilitate the training of a new model and the implementation of transfer learning by freezing the majority of neural network layers and retaining the last two Dense layers of the eViT[54]. This refinement of an existing eViT model reduces the time required for training and the labeling of data[34]. Additionally, it can be trained on a central processing unit (CPU) rather than a graphics processing units (GPU).

IVEA shows robust analysis capability even when the noise power is equal to that of true events by extracting spatiotemporal features of the signal (Fig. 2c). Additionally, IVEA is capable of discarding artifacts such as short focus changes resulting in transient signal variation. However, slow focus drift cannot be compensated as well as lateral drift in the case of stationary burst events. Upon drift an active synapse is detected at different locations and is assigned to several coordinates. Therefore, drift should be corrected using free plugins such as but not limited to NanoJ core[55] prior to IVEA analysis. Random burst event does not require drift correction as the algorithm does not require fixed spatial coordinates. Finally, to detect exocytosis events, the IVEA algorithm requires analyzing the first four frames from each video. This step is important for the automation process, so that IVEA can learn from the images that are devoid of events (see method section "stationary and random burst events algorithm"). In rare cases in which videos contain events within these frames, IVEA can generate learning parameters that enable the detection of high SNR events while events with low SNR may be overlooked. However, manually reducing the detection threshold to 1 or less can enhance the sensitivity of event selection. The neural network can classify these increased detections correctly, while eliminating most of the FP events. As a result, a heightened computational time requirement may arise. This can be mitigated by extracting more than four frames that are devoid of events and adding them to the beginning of the movie.

In conclusion, IVEA reduces analysis time by >90%, requiring minimal to no human input, and significantly decreases analysis bias. Furthermore, the number of fusion events detected by IVEA is higher than those detected by the human expert, especially when it comes to visualizing rare, i.e. low brightness, events in low-SNR conditions. This allows us to generate large datasets with meaningful statistical power. In addition, IVEA unlike the human expert readily detects short events, which not only increases the number of detected events but also promotes the understanding of biological processes. The earlier application of hotspot area extraction module for AndromeDA revealed not only discrete dopamine (DA) release and extracellular DA diffusion, but it also enabled the discovery of heterogeneous release hotspots - albeit using a less-advanced algorithm at the time. The trove of information collected from individual events allowed to elucidate the role of key proteins in the molecular machinery of exocytosis in dopaminergic neurons[39].

We foresee that our program will be adaptable to analyze exocytosis in many other systems and to extend its use to other burst-like events. For instance, the eViT method with additional training was proficient at analyzing transient localized calcium signals with low SNR, such as calcium sparks. The "hotspot area extraction" should be capable of analyzing calcium waves in cell clusters, brain slices or in vivo measurements. Using IVEA for detecting exocytosis events has the potential to rapidly advance research in the neuroscience, immunology, and endocrinology.

## Methods
### Stationary and random burst events algorithm
Our approach to detecting burst events involves two main parts: automatic region selection and neural network classification. The first part is detecting the potential ROIs. The event detection algorithm used by IVEA employs the grayscale image foreground detection

method, which leverages a bidirectional image subtraction technique. This involves subtracting the intensity values in a 16-bit image stack of a reference frame $I_i$ from an offset frame $I_{i+n}$ in both directions—forward ($\Delta I_F = I_{i+n} - I_i$) and backward ($\Delta I_B = I_i - I_{i+n}$). This approach simplifies the handling of 32-bit matrices, reducing potential complexities with straightforward data processing. Backward subtraction is employed only with random burst events to detect fusion events visualized with a pH-insensitive stain. In the absence of fluctuations or foreground variations, the subtraction results in an image with pixel values of zero. However, the presence of noise and/or artifactual fluctuations during image acquisition makes it difficult to differentiate real events from artifacts. To distinguish real events from noise, we detect local maxima (LM) in the subtracted images, representing potential exocytic hotspots. Subsequently, an ROI is generated around each identified LM. These ROIs are then employed to generate image patches, which correspond to cropped sections of video frames. This approach captures localized activities over time, thereby enabling the isolation of specific events for classification rather than analyzing the entire video frame-by-frame with the neural network. The LM detection and ROI extraction process is fully automated, incorporating a global thresholding step that learns from the first four frames of the processed video. This ensures that noise and spurious maxima are filtered out, leaving only meaningful events for classification. The local minima prominence $\rho$ approximation algorithm iterates over the noisy images; each iteration increments the value $\rho_n$ until the number of LM coordinates ($l_n$) equals 0. If, instead, four successive iterations yield the same number of maxima (e.g., $l_n = l_{n-4}$), the program sets $\rho = \rho_n$ where "$n$" is the iteration number. These four images are also utilized to estimate the full width at half maximum (FWHM) $\sigma$ of the noise LMs, and to measure $\Delta\mu$, which is the average of the mean intensities $\Delta\mu_j$ at LM $C_j$ with radius $r$ expressed as:

$$\Delta\mu_j = \frac{1}{4r^2+1} \sum_{C_j^{(x)}-r}^{C_j^{(x)}+r} \sum_{C_j^{(y)}-r}^{C_j^{(y)}+r} \Delta I_i(x_j, y_j) \, with \, r \in \mathbb{N} \tag{1}$$

The region selection procedure is similar to parameter automation. We determine $\Delta\mu_j$ and $\sigma_j$ for each event $E_j$ at $C_j$. To designate $E_j$ as a selected region we employ the following condition:

$$E_j \left| \left( \Delta\mu_j > \Delta\mu \cdot \theta \right) \wedge \left( \sigma_j > \sigma \right) with \, \sigma, \theta \in \mathbb{R} \right. \tag{2}$$

Here, θ denotes the sensitivity parameter, which can be adjusted by the user.

After detection, IVEA performs spatiotemporal tracking for ROI recognition and labeling. This is applied for each detected ROI coordinate over a certain radius and period. For burst events exhibiting temporal dynamics (e.g., lytic granule fusion), a sequence of image patches is extracted, encompassing frames both preceding and following the time point of the event. Each sequence is fed to a shared encoder layer attached to the ViT architecture for image recognition as described in Dosovitskiy et al.[36]. We designated the modified architecture as eViT. The image patches' spatial dimensions are variable, but they are scaled to fit the encoder input layer of $32 \times 32$ dimensions. Each patch represents the extracted area centered at the LM, while the sequence is centered on the fluorescence intensity peak time (Fig. 1a). The encoder network automatically extracts features from each sequence and forwards the encoded data to a multi-layered perceptron (MLP), which in turn forwards the data to the ViT network. The ViT network then performs positional encoding on the extracted features and classifies each sequence as a true event or not (Fig. 1b).

For stationary burst events, we employed a straightforward model architecture comprising an LSTM network[37] for exocytosis classification. Our LSTM architecture is designed for multivariate time series classification[49] (Fig. 1c). In this case, the image patches undergo feature

extraction preprocessing to convert them into one-dimensional time-series vectors (Fig. 1a). These feature vectors are subsequently fed into the LSTM network for classification (Fig. 1d). An additional optional method is implemented in the stationary burst events for detecting and tracking agonist/electric and $NH_4^+$ stimulations. This algorithm is utilized to recognize and sort events based on their occurrence period. Stimulus detection is expressed as:

$$\mathcal{R}_i = \frac{\Delta\mu_{i+1}}{\Delta\mu_i} > \theta_s \tag{3}$$

Where $\mathcal{R}_i$ is the mean ratio; $\theta_s$ is the default threshold, is 1.1; $\Delta\mu_i$ and $\Delta\mu_{i+1}$ are the mean gray value of image $\Delta I_i$ and $\Delta I_{i+1}$ respectively.

To avoid an increase in the number of detected events due to high fluorescence intensity during the stimulus period, we adjust the detection sensitivity by increasing the detection sensitivity $\theta_i$, such as $\theta_i = \theta \cdot \mathcal{R}_i$.

## Feature extraction

Feature extraction is performed by extracting a sequence of image patches around $C_j$ over a time interval. This patch is subdivided into smaller regions. Subsequently, we determine the mean intensity of each region (Fig. 1a). For each selected region denoted as $E_j$, we extract the spatial neighboring pixels around $C_j$ as a 2D matrix $\mathbf{M}^j \in \mathbb{R}^{k \times k}$, where $k$ is the kernel defined by the user. The spatiotemporal data $\mathbf{V}_j \in \mathbb{R}^{k \times k \times T}$, which represents event $E_j$ that occurred at time $t_j$, is extracted over several frames T expressed as (3).

$$\mathbf{V}_j(x, y, t) := \left\{ \mathbf{M}_t^j(x, y) \,\middle|\, t \in \left[ t_j - n_b, t_j + n_a \right] \right\} \tag{4}$$

Whereas $n_b$ is the number of frames before $t_j$ and $n_a$ is the number of frames after $t_j$. The spatial coordinates of each matrix $\mathbf{M}^j$ are split into 13 small regions $f \in \mathbb{N}$ (see Supplementary Fig. 13a), which yield the feature matrix $\mathbf{P}_j \in \mathbb{R}^{T \times f}$ for event $E_j$ such as (4):

$$\mathbf{P}_j(\mathbf{x}(t), f) := \left\{ \frac{1}{n_f} \sum_{f=1}^{n_f} \mathbf{V}_j\left( x_f, y_f, t \right) \,\middle|\, f \in [1, f] \right\} \tag{5}$$

Whereas $n_f$ is the number of pixels in each region $f$.

This approach forms time series data $\mathbf{P}$, represented as $\mathbf{P} \in \mathbb{R}^{n_s \times T \times f}$, where $n_s$ is the total number of nominated events. Each element $\mathbf{P}_j$ in $\mathbf{P}$ represents 13 distinct signals that capture the fluorescence intensity profile of different regions plotted on a single graph (Fig. 1d, Supplementary Fig. 9). The choice of small, symmetrical regions over circular masks enhances feature preservation of the spatiotemporal signal (Supplementary Fig. 13a, c). Additionally, we opted for 13 regions over 9 regions due to their higher sensitivity in capturing slow granule movements (Supplementary Fig. 13a, b).

If the number of frames increases by $t_n$, such as $\mathbf{x}'(t) \in \mathbb{R}^\ell$ with $\ell = T + t_n$. $\mathbf{P}(\mathbf{x}(t), f) \in \mathbb{R}^{T \times f}$ is then determined by using windowing mean-sampling expressed as:

$$\mathbf{x}(t)_i = \left\{ \frac{1}{\omega} \sum_{k=1}^{\omega} \mathbf{x}'(t)_{\omega(i-1)+k} \,\middle|\, \omega = \frac{\ell}{T} \, with \, \omega \in \mathbb{N} \right\} \tag{6}$$

Where, $\mathbf{x}(t)_i$ is the $i-$th element of the sampled vector $\mathbf{x}(t)$, and $\mathbf{x}'(t)_j$ as $j = \omega(i-1) + k$ is the $j-$th element of vector $\mathbf{x}'(t)$.

For stationary burst events, we used $n_b = 10$ and $n_a = 30$ time samples, resulting in a total of $T = 41$ time samples. For random burst events, the implemented LSTM model used a total number of time samples $T = 21$ ($n_b = 10$ and $n_a = 10$). The time series data $\mathbf{P}$ is first normalized within the range of 0 to 1 before being converted to tensors. For stationary burst events analysis, $\mathbf{P}$ is converted into 3D

tensors $\mathcal{T} \in \mathbb{R}^{n_s \times \mathfrak{f} \times T}$ as follows:

$$\mathbf{P}(j, \mathbf{x}(t), \mathfrak{f}) \rightarrow \mathcal{T}(j, \mathfrak{f}, \mathbf{x}(t)) \,|\, j \in [1, n_s] \, with \, n_s \,, j \in \mathbb{N} \qquad (7)$$

## Multivariate LSTM neural network architecture

Our LSTM network comprises four different layers. It serves as a robust framework for multivariate temporal data analysis. The first input layer, defined by the input-shape specification, establishes the dimensions of the incoming multivariate time series data. This initial stage isn't a distinct processing layer but rather a configuration step to align the network with the input data's structure. The subsequent architecture unfolds with a 1D convolution layer employing Rectified Linear Unit (ReLU) activation function. The subsequent layer incorporates a LSTM, designed to recognize sequential patterns. To promote stable training dynamics, a batch-normalization layer is added. The last layer is the fully-connected Dense layer that employs the softmax activation function, rendering the architecture adept for multiclass classification.

$$\hat{y}_i = h(z) = \frac{e^{z_i}}{\sum_{j=1}^{n_c} e^{z_j}} \qquad (8)$$

Here, $h(z)$ is the softmax function, $z_i$ represents the raw score (logit) for a specific class $i$ and $n_c$ is the number of classes. This arrangement encapsulates both localized and temporal patterns inherent to multivariate sequential data, combining convolution, recurrent, and normalization mechanisms. The network is structured to accommodate categorical cross-entropy as the loss function $\mathcal{L}$ (Eq. 9), tailored for multiclass categorization, while optimization leverages the Adam optimizer with a learning rate of $3 \times 10^{-4}$.

$$\mathcal{L} = \frac{1}{n_s} \sum_{j=1}^{n_s} L(\hat{y}^{(j)}), \, with \, L(\hat{y}^{(j)}) = - \sum_{i=1}^{n_c} y_i^{(j)} \cdot \log(\hat{y}_i^{(j)}) \qquad (9)$$

Where $L(\hat{y}^{(j)})$ is the loss for a single data point (sample), $n_s$ is the number of data points, $y_i \in \{0, 1\}$ is the true label for class $i$ and $\hat{y}_i$ is the predicted probability.

## Encoder-ViT network architecture

Our eViT network consists of two components: a CNN as the encoder for feature extraction from image patches and a ViT for classification. The encoder is shared CNN. It comprises seven layers, including a 2D spatial convolution layer that is followed by a sequence of 3D convolution layers and 3D max pooling operations (Supplementary Fig. 14a, b).

We have two pre-trained models available with IVEA, namely GranuVision2 and GranuVision3. The model's encoder input layer accepts time series image patches of length 26 for GranuVision2 and 28 for GranuVision3. The width, height and channel of each patch is $32 \times 32 \times 1$, denoted as $\mathbf{X} \in \mathbb{R}^{t \times w \times h \times c}$. If the dimensions of the image patches change, we use bilinear interpolation to resize the images to $32 \times 32$. The initial stage of the shared CNN involves the application of a 3D convolution layer with 16 filters and ReLU activation, which is followed by a 3D max pooling layer. Subsequently, another 3D convolution layer with 32 filters and ReLU activation is applied, followed by a 3D max pooling layer. Finally, a 3D convolution layer with 64 filters and ReLU activation is applied, followed by a 3D max pooling layer. The output of the encoder is passed through a Flatten layer and a 64-unit Dense layer. Positional embeddings are added before the sequence enters the transformer block, which includes an MLP. The transformer block consists of multi-head self-attention and an MLP, each with a residual connection (Fig. 1b). The transformer block operates on an input sequence of length equal to the time series dimension, with a key dimension representing the output size of the previous Dense layer.

The MLP inside the transformer block consists of two Dense layers, where the first layer has a dimension twice the key dimension, and the second layer projects it back to the key dimension. The final MLP comprises two Dense layers: the first layer with GELU non-linearity, and the second with SoftMax activation, classifying 10 (2 fusion + 8 artifacts) or 11 (3 fusion + 8 artifacts) classes for GranuVision2 or GranuVision3, respectively. The eViT architecture underwent ablation to study the impact of layers on model performance. This study involved probing the model on an evaluation dataset by eliminating layers and retraining. The evaluation dataset was divided into two categories: exocytosis (positive labels) or non-exocytosis (negative labels). We performed the ablation study on the shared CNN layers and the penultimate Dense layer. The results show that progressively removing these layers leads to a noticeable decline in performance. The final configuration, with only one convolution layer, exhibited the strongest performance drop. This study underscores the additive role of each layer in our current eViT architecture (see Supplementary Fig. 15).

## Gaussian non-max suppression

Various non-maximum suppression techniques are typically used to address the multiple overlapping-detections problem, including the classical intersection over union (IoU) method, weighted boxes fusion (WBF), and others[56]. However, these methods cannot be implemented with our data because exocytotic events cannot be limited to objects with boundaries or boxes. Thus, we have developed a new method that implements 3D Gaussian spread over time

$$g(x, y, t) = \Delta\mu \cdot \exp\left(-\frac{(\Delta x^2 + \Delta y^2)}{2(\sigma + SR)^2}\right) \cdot \exp\left(-\frac{\Delta t^2}{2\tau^2}\right) with \, \tau = \nu \cdot \sigma \quad (10)$$

where $\Delta\mu$ is the fluorescence intensity of the event area in the subtracted image; $\sigma$ is the event's cloud spread; SR = 1 is a user-controlled spread radius; $\tau$ is the temporal cloud spread; and $\nu$ is the image acquisition frequency set to 10 Hz.

To isolate prime events from redundant ones, we apply $g(x, y, t)$ to each pair of events as follows:

$$\begin{cases} E_i = E_j & if \, \Delta\mu_i < g_j(x_j, y_j, t_j) \\ E_i \neq E_j & otherwise \end{cases} \qquad (11)$$

where $E_i$ and $E_j$ are two different true positive (TP) events.

## Neural network training and evaluation

Python was utilized for developing and training the LSTM and eViT networks, while Visual Studio Code was employed for coding. The LSTM training data was processed and prepared in MATLAB, enabling visualization of patterns in segmented regions of time series image patches. In contrast, the eViT network's data labeling was managed using ImageJ. The initial data labeling and IVEA classification validation were performed by the human experts listed in Supplementary Table 2.

For the eViT, the videos were associated with their corresponding ROI files. These files include ROI center coordinates, frame positions, and radii. The IVEA software was used to export the labeled ROIs as zip files, with the training datasets being tagged as _training_rois to differentiate them from the evaluation data. The user can enable this option in the IVEA graphical user interface (GUI) via the Select Model dropdown list. Prior to integrating the neural network with IVEA, the initial procedure involved exporting the selected regions identified by the automation processes and manually labeling them. Subsequent to the initial integration of the neural model, events in the training datasets were automatically labeled with a uniform naming convention that includes the list number, event ID, frame number, and classification category. For example, the third event in the ROI manager list,

with an ID of 3, detected at frame 779 and initially classified as category 1, would be named: 3-event (3) | frame 779_class_1. To prepare the data for the neural network training, we have developed a Python-based script with a simple interface and a JSON configuration file to extract or load the training data. The process of extracting the training data entails the following two steps. First, the ImageJ ROIs are read to identify the events positions and their categories. Subsequently, the times series patches are extracted at each ROI. These data are then stored as Hierarchical Data Format (.h5) file format, organized into dictionaries containing x_train and y_train data, facilitating efficient loading and archiving of the training data.

During the training process, labels were refined through an iterative process. Initially, the network was trained to distinguish between exocytosis and not exocytosis. Later on, additional classes were introduced to differentiate between exocytosis subtypes, as well as motion or noise artifacts. Exocytosis classes received positive integers (e.g., 0, 1, 2), while non-exocytosis classes (such as noise or artifacts) were assigned negative integers (e.g., -1, -2). Whenever a misclassification was noted (for instance, class_1 instead of class_2 or -6), the label was corrected or a new one was defined, and the data were fed back into the network for retraining. To accommodate the substantial volume of events predicted and labeled by IVEA, a significant number of labels associated with non-exocytosis events were excluded to enhance data management. For generating a new model, training files and tools are available on our GitHub page for IVEA (see data available section).

For the LSTM, the data were exported as CSV files in the form of $\mathbf{X} \in \mathbb{R}^{n_s \times f \times t}$ for stationary burst events, and the labeled data as $\mathbf{Y} \in \mathbb{R}^{n_s \times n_c}$, where, $n_s$ is the number of samples and $n_c$ is the number of classes. Our LSTM network for stationary burst events was trained with 39 videos with 11,300 data samples with dimensions of 13 ×41, while for the random burst events the LSTM network was trained on 548 videos with 12,600 data samples. As for the eViT network, the data were saved as videos with their associated ROI files both in zip and roi file container. The input data for the eViT network is $\mathbf{X} \in \mathbb{R}^{n_s \times t \times w \times h \times c}$ and $\mathbf{Y} \in \mathbb{R}^{n_s \times n_c}$. The eViT network for random burst events was trained with 608 videos and 7931 data samples augmented to reach 24,916 samples of 26 x 32 x 32 dimensions each (see Supplementary Table 7). These videos were acquired at a rate of 10 Hz. For videos in which the eViT was tested and acquired at 50 Hz, the videos were reduced by a factor of five using the ImageJ "reduce" function, resulting in a rate of 10 Hz. We used an NVIDIA RTX 3070 to training the neural network.

The final models were evaluated on videos unseen by the neural network, rather than reserving part of the training data for validation. A diverse array of datasets was utilized in the evaluation process, acquired from multiple laboratories employing a variety of microscopes. These included lytic granule exocytosis in T cells, dense-core granules in INS-1 and chromaffin cells (both pH-sensitive and pH-insensitive fluorescent markers), DRG neurons expressing Synaptophysin-SEP, and dopaminergic neurons examined with dopamine nanosensors ("AndromeDA"). The analysis showed strong concordance with HE annotations. Most of the datasets were processed using default IVEA settings and automated parameter estimation, though users may override these defaults when necessary. For parameter estimation, these data sets are devoid of fusion events that occurred within the initial four frames of the videos, enabling IVEA to learn from. However, users have the option to disable automated learning and opt for manual override by adjusting the sensitivity to 1 or lower, thereby generating additional local maxima coordinates.

For each unseen video analyzed by IVEA, the resulting event ROIs were examined for validation. HE labels every detected event as true exocytosis (true positive, TP) or falsely predicted (false positive, FP). Any missed exocytosis event that was previously observed by the HE was labeled as a false negative (FN). Finally, precision, recall, and

F1 score were computed based on the TP, FP, and FN counts, with the corresponding formulas:

$$\text{Precision} = \frac{TP}{TP + FP} \tag{12}$$

$$\text{Recall} = \frac{TP}{TP + FN} \tag{13}$$

$$\text{F1 Score} = 2 \times \frac{Precision \times Recall}{Precision + Recall} \tag{14}$$

The IVEA analysis was conducted on a range of computer systems, with a baseline configuration of an Intel Core i5 processor and 32 GB of RAM without GPU.

## Training on new data
Training the neural network on new data involves two steps: data labeling and neural network training. To label the data, the user can create an ROI over the event using the ImageJ "ROI Manager" tool. The user should then label the ROIs with a special tag and with their associated category number, and save the ROI/s as a roi/zip file under the same name as the video. Alternatively, users can employ the IVEA "Data labeling" ImageJ plugin, which is provided with IVEA for easy labeling. The next step involves training the neural network using Python. Users should set up a Python development environment, such as the Anaconda platform. To run the IVEA training GUI, users must install the libraries associated with the Google TensorFlow platform. The script launches a GUI that enables users to combine their labeled data with an existing dataset, collect a new dataset, train a new neural network, or refine an existing one. After training, the script generates a trained Keras model and saves it to the designated directory with its associated JSON configuration file. The IVEA plugin enables users to import a custom model for subsequent predictions. If no custom model is imported, IVEA uses its embedded models.

## Google TensorFlow-Java implementation
IVEA's LSTM and eViT networks were developed using the Python v3.8.15 language and Google's machine learning and artificial intelligence framework TensorFlow v2.9.1 or v2.10 with the Keras library. Using Python, we were able to train our neural network and export the trained model as a Protocol Buffer (.pb) file. To load and use our model with ImageJ Fiji, we used the Google TensorFlow Java v1.15.0 library and deeplearning4j core v1.0.0-M1.1 in our software. Integrating Google TensorFlow with Java is a complex task, particularly within the context of Fiji implementation. While Java offers versatility, it has limitations compared to Python, particularly in providing user-friendly and adaptable tools for machine learning applications. Notably, the Java support for Google TensorFlow is constrained, and as of the year 2024, faced issues with deprecated documentation. Additionally, the consolidation of all components into a single Java archive (jar) file poses challenges within the Fiji environment. In an effort to simplify the integration of the Google Deep Learning Framework with Java inside Fiji, we provided a concise explanation of the TensorFlow Java implementation on https://github.com/AbedChouaib/IVEA.

## Video simulation and noise control
To mimic the CTL's lytic granules and simulate the fusion activity, we first create the vesicles as small spheres with Gaussian intensity spread with a cutoff at $2\sigma$ using equation:

$$g(x,y) = \mu \cdot \exp\left(-\frac{\left((x - x_c)^2 + (y - y_c)^2\right)}{2(\sigma)^2}\right) \tag{15}$$

Whereas $\mu$ is the intensity of the spot and $\sigma \in [1.1, 3]$, $\sigma \in \mathbb{R}$ is the standard deviations controlling the spatial spread distribution.

We then added some random spatial movement for the vesicles to add motion variable. The vesicle's fusion was more like fluorescence intensity cloud that spreads and disappears. To simulate these phenomena, we used Gaussian spread over time equation to control the temporal presence of the fusion over time, then we added one more variable for the radial spatial spreading dependent on the time variation. The overall equation expressed as:

$$h(x,y,t) = \mu \cdot \exp\left(-\frac{\Psi(x,y,t)}{(2\sigma_s)^2}\right) \cdot \exp\left(-\frac{(T-t)^2}{(2\tau)^2}\right) \quad (16)$$

$$\Psi(x,y,t) = (\Delta x^2 + \Delta y^2) \cdot \exp\left(\frac{(T-t)^2}{(2\tau)^2}\right) \text{ with } \sigma_S, \tau > 0 \quad (17)$$

Whereas $\Psi(x,y,t)$ simulate the radial dynamic dispersion of vesicle cargo over time, $t$ is the current frame, $T$ is the frame where the fusion occurs, $\tau$ is the fusion time interval, $\sigma_S$ is the fusion radial spread and $\mu$ is the fluorescence intensity magnitude.

For the noise control analysis, twenty videos with distinct SNRs were generated using MATLAB. Initially, a baseline video with no noise was created. Artificial white noise was then added to this baseline using the built-in function imnoise(). Subsequently, the MATLAB built-in poissrnd() function was utilized to generate random photon shot noise commonly observed in microscopy. A Gaussian blur was applied to it to replicate the point spread function seen in microscopy images. Finally, the processed noise was added to the video to achieve the desired noise characteristics. The Poisson noise was modeled using the following equation:

$$I_{noise}(x,y) := \text{Poisson}(\lambda \cdot I(x,y)) \quad (18)$$

Where $\lambda$ is the scaling factor that controls the relative level of noise.

To explore the impact of noise across a range of conditions, the scaling factor $\lambda$ was varied incrementally from 0.1 up to 10 times the signal. Higher values of $\lambda$ correspond to higher noise levels (lower SNR), while lower values of $\lambda$ reduce the noise relative to the signal (higher SNR).

## Hotspot area detection algorithm

The IVEA hotspot area extraction is based on DART algorithms, which employ unsupervised learning to segment the image into different layers. Following image segmentation, the MIC algorithm is performed to address the non-uniform regional fluctuations in fluorescence intensity, which is conducted prior to foreground subtraction. MIC is an enhanced version of the simple ratio and the previous method used with DART. MIC clusters the first image into a series of layers, wherein each layer comprises a group of labeled pixels that exhibit a close range of gray values, as determined by k-mean clustering. This process can be expressed as $I(x,y) \rightarrow I(x,y,k)| k = \mathbb{N}^5$ (Supplementary Fig. 12). Conventional approach (DART) involves the addition of the difference in gray values of clusters between two subsequent images. In contrast, with MIC we employed a simple ratio equation for each layer, assuming that the least cluster value represents the background. In the event of uniform regional fluctuations, the number of clusters of MIC could be reduced to $k = 1$; this would yield a result similar to that of the simple ratio. MIC is expressed as:

$$I_i'(k,x,y) := \left(\left(\frac{\mu_{i-n}(k)}{\mu_i(k)} - 1\right) \cdot \theta + 1\right) \cdot I_i(k,x,y) \quad (19)$$

Where $i$ is the $i$-th frame, $k$ is the number of layers, $n$ is the frame difference, and $\theta$ is a user input parameter added to control intensity adjustment, default $\theta = 1$.

The iterative threshold consists of two distinct parts: Initially we capture two images where no events have occurred. Next, we compute $\Delta I$ and transform it into an 8-bit image to decrease computation time by reducing the iterations to under 255 steps. Finally, we attempt to clear $\Delta I$ repeatedly. The clearing process consists of three sequential operations: threshold, erosion, and median filtering. In the iterative process, the threshold starts at half the mean intensity of $\Delta I$, then we perform erosion with kernel $K_e[n,n]$ to eliminate lone pixels $\Delta I = \Delta I \ominus K_e$ as $n = 3$. After erosion, a median filter with a user-defined radius or a preset default value is applied. The average mean gray value of the processed image is calculated and checked to see if it is equal to zero. If not, iteratively the threshold increments by one gray step until we reach an average mean value of zero. The outcome of this process delivers the first iterative threshold decision $v_1$. The second threshold decision $v_2$ is performed for the remaining images, where this threshold is determined to correct the first threshold. The second threshold is like the previous process, except that a specific area of the segmented background is selected from each image (Fig. 5b). The final threshold decision $v_i$ is determined by $v_i = v_2 \cdot \alpha$ where $\alpha$ is the threshold sensitivity, if $\alpha$ was set as zero. The software takes two more frames to learn the sensitivity; it assumes no events had occurred and tries to correct $v_i$ by tuning $\alpha$ automatically. This step adjusts the difference between iterating over the entire image and iterating exclusively over the image's background. Regions surpassing the global threshold $v_i$ are considered as detected occurrences. Subsequently, each contiguous region is isolated and assigned a distinctive label designating it as an event. The fluorescence intensity of each event is spatially and temporally tracked immediately after detection. The mean intensity of each event $\mu_e(t)$ is temporally measured over a fixed area, then we determine the mid-intensity $\mu_{mid} = \frac{1}{2}(\mu_e(t_{min}) + \mu_e(t_{max}))$. When the fluorescence intensity of an event falls below $\mu_{mid}$, the event signal is considered to have disappeared and the tracking stops (Fig. 5c,d).

## Mice for T Cell, chromaffin cell and DRG neuron culture

WT mice with C57BL/6 N background used in this study were purchased from Charles River. Synaptobrevin2-mRFP knock-in (KI) mice were generated as described in Matti et al.[14]. Granzyme B-mTFP KI mice with C57BL/6 J background were generated as described previously[57]. Granzyme B-tdTomato KI mice[58] were purchased from the Transgenesis and Archiving of Animal Models (TAAM) (National Centre of Scientific Research (CNRS), Orleans, France). Mice were housed in individually ventilated cages under specific pathogen-free conditions in a 12 h light-dark cycle with constant access to food and water. All experimental procedures were approved and performed according to the regulations by the state of Saarland (Landesamt für Verbraucherschutz, AZ.: 2.4.1.1).

## Murine CD8 + T cells

**Culture.** Splenocytes were isolated from 8–20 week-old mice of either sex as described before[22]. Briefly, naive CD8 T cells were positively isolated from splenocytes using Dynabeads FlowComp Mouse CD8+ kit (Thermo Fisher Scientific, Cat# 11462D) as described by the manufacturer. The isolated naive CD8 + T cells were stimulated with anti-CD3ε /anti-CD28 activator beads (1:0.8 ratio, Thermo Fisher Scientific, Cat# 11453D) and cultured for 5 days at 37 °C with 5% CO$_2$. Cells were cultured at a density of $1 \times 10^6$ cells/ml in 12 well plates with AIMV medium (Thermo Fisher Scientific, Cat# 12055083) containing 10% FCS (Thermo Fisher Scientific, Cat# A5256901), 50 U/ml penicillin, 50 µg/ml streptomycin (Thermo Fisher Scientific, Cat# 15140163), 30 U/ml recombinant IL-2 (Thermo Fisher Scientific, Cat# 212-12-100 µg) and

50 µM 2-mercaptoethanol (Sigma, Cat# M6250). Beads and IL-2 were removed from T cell culture 1 day before experiments.

**Transfection and constructs.** Day 4 effector T cells were transfected 12 h prior to the experiment through electroporation of the Plasmid DNA (Granzyme B-pHuji, Synaptobrevin2-pHuji, CD63-pHuji) using Nucleofector™ 2b Device (Lonza) and the nucleofection kit for primary mouse T cells (Lonza, Cat# VPA-1006), according to the manufacturer's protocol (Lonza). After nucleofection, cells were maintained in a recovery medium as described by Alawar et al.[59]. 4 h prior to the experiment the cells were washed with AIMV medium. The pMax-granzyme B-pHuji construct was generated by replacing the mTFP at the C-terminus of pMax-granzyme-mTFP[57] with pHuji using a forward primer that included an AgeI restriction site 5′-ATG TAT ATC CAC CGG TCG CCA CCA TGG TGA GCA AGG GCG AGG AG-3′ and a reverse primer that included a NheI restriction site 5′-ATG TAT AGC TAG CTT ACT TGT ACA GCT C-3′. The size of this plasmid was 4.315 kb. The pmax-CD63-pHuji was generated by subcloning from pCMV-CD63-pHuji[60], which was a generous gift from Frederik Verweij (Centre de Psychiatrie et neurosciences, Amsterdam/Paris), into pMax with the restriction sites EcoRI and XbaI. Its size was 4.282 kb. Synaptobrevin2-pHuji plasmid was generated as described in ref. [61].

**Acquisition conditions.** Measurement of exocytosis was performed via TIRFM as follows. We used day 5 bead activated CTLs isolated from GzmB-mTFP KI, GzmB-tdTomato KI, Synaptobrevin2-mRFP KI or WT mice. The latter were transfected with the above described constructs. $3 \times 10^5$ cells were resuspended in 30 µl of extracellular buffer (10 mM glucose, 5 mM HEPES, 155 mM NaCl, 4.5 mM KCl, and 2 mM $MgCl_2$) and allowed to settle for 1–2 min on anti-CD3ε antibody (30 µg/ml, BD Pharmingen, clone 145-2C11) coated coverslips to allow immunological synapse formation triggering lytic granule exocytosis. Cells were then perfused with extracellular buffer containing calcium (10 mM glucose, 5 mM HEPES, 140 mM NaCl, 4.5 mM KCl, 2 mM $MgCl_2$ and 10 mM $CaCl_2$) to stimulate CG secretion. Cells were recorded for 10 min at $20 \pm 2\,°C$.

**Imaging.** Live cell imaging was done with two setups. The experiments performed with CTL (lytic granule staining with synaptobrevin-mRFP, granzyme B-mTFP or granzyme B-tdTomato) were performed with setup # 1 described previously[22,26,45]. Briefly, an Olympus IX70 microscope (Olympus, Hamburg, Germany) was equipped with a 100x/1.45 NA Plan Apochromat Olympus objective (Olympus, Hamburg, Germany), a TILL-total internal reflection fluorescence (TILL-TIRF) condenser (TILL Photonics, Kaufbeuren, Germany), and a QuantEM 512SC camera (Photometrics, Tucson, AZ, USA) or Prime 95 B scientific CMOS camera (Teledyne Photometrics, Tucson, AZ, USA). The final pixel size was 160 nm and 110 nm, respectively. A multi-band argon laser (Spectra-Physics, Stahnsdorf, Germany) emitting at 488 nm was used to excite mTFP fluorescence, and a solid-state laser 85 YCA emitting at 561 nm (Melles Griot Laser Group, Carlsbad, CA, USA) was used to excite mRFP and tdTomato. The setup was controlled by Visiview software (Version:4.0.0.11, Visitron GmbH). The acquisition frequency was 10 Hz for all experiments.

The setup # 2 used to acquire CTL secretion, in which the lytic granules were labeled by Synaptobrevin-pHuji, granzyme B-pHuji and CD63-pHuji overexpression, was previously described[57,61]. Briefly the setup from Visitron Systems GmbH (Puchheim, Germany) was based on an IX83 (Olympus) equipped with the Olympus autofocus module, a UAPON100XOTIRF NA 1.49 objective (Olympus), a 445 nm laser (100 mW), a 488 nm laser (100 mW) and a solid-state 561 nm laser (100 mW, Melles Griot Laser Group, Carlsbad, CA, USA). The TIRFM angle was controlled by the iLAS2 illumination control system (Roper Scientific SAS, France). Images were acquired with a QuantEM 512SC camera (Photometrics, Tucson, AZ, USA) or Prime 95 B scientific CMOS camera (Teledyne Photometrics, Tucson, AZ, USA). The final pixel size was 160 nm and 110 nm, respectively. The setup was controlled by Visiview software (Version 4.0.0.11, Visitron GmbH). The acquisition frequency was 5 or 10 Hz, and the acquisition time was 10 to 15 min.

## Murine DRG neurons
**Culture and transfection.** The training of the Stationary burst event neural network and the automatic detection of neuronal exocytosis at synapse was performed on data sets that were previously published[45,46]. Shortly DRG neuron cultures from young adult (1–4 weeks old) WT of either sex was made as previously described[26]. Lentivirus infection to transfect with SypHy was performed on DIV1. The following day, the lentivirus was removed by washing before adding the second order spinal cord (SC) interneurons (SC neurons) to the culture to allow DRG neurons to form synapses. SC neurons were prepared from WT P0-P2 pups of either sex using as previously described[45]. DRG/SC co-culture was maintained in Neurobasal A (NBA) medium (Cat#° 21103049) supplemented with fetal calf serum (5% v/v, Cat# 11550356), penicillin and streptomycin (0.2% each, Cat# 11548876), B27 supplement (2%, Cat# 17504-044), GlutaMAX (1%, Cat# 35050-061, all products from Thermo Fisher Scientific, Waltham, MA, USA), and human beta-nerve growth factor (0.2 µg/mL, Cat# N245, Alomone Labs, Jerusalem, Israel) at 37 °C and 5% $CO_2$.

**Acquisition conditions.** Secretion was evoked by electrical stimulation via a bipolar platinum-iridium field electrode (Cat# PI2ST30.5B10, MicroProbes, Gaithersburg, MD, USA) and a pulse stimulator (Isolated Pulse Stimulator Model 2100, A-M Systems, Sequim, WA, USA). The measurement protocol was 30 s without stimulus followed by a biphasic 1 ms long 4 V stimulus train at 10 Hz for 30 s to elicit exocytosis of SVs. At the end of the measurement, $NH_4Cl$ was applied to visualize the entire SV pool. During the measurement, the temperature was maintained at 32 °C by a perfusion system with an inline solution heater (Warner Instruments, Holliston, MA, USA). The extracellular solution contained 147 mM NaCl, 2.4 mM KCl, 2.5 mM $CaCl_2$, 1.2 mM $MgCl_2$, 10 mM HEPES, and 10 mM glucose (pH 7.4; 300 mOsm). The $NH_4Cl$ solution had the same composition as the extracellular solution, but the NaCl was replaced with 40 mM $NH_4Cl$. All products were from Sigma-Aldrich/Merck.

**Imaging.** All experiments were performed on Setup # 1 described above for the CTLs.

## Chromaffin cells
Data showing bovine chromaffin cells exocytosis was from Becherer et al.[62] and Hugo et al.[14]. Culture condition was described by Ashery et al.[63]. Briefly, chromaffin cells were dissociated from the bovine adrenal gland by enzymatic dissociation (20 min) with 129.5 units per ml collagenase (Cat# C1-22, Biochrom AG, Berlin Germany). They were maintained for 3–5 days in culture in DMEM (Cat# 31966021) containing ITS-X (1:100 dilution, Cat# 51500056), Penicillin/Streptavidin (1:250, Cat# 15070063) all products from Thermo Fisher Scientific, Waltham, MA, USA. They were electroporated with NPY-mRFP to label the large dense core granules using the Gene Pulser II (Biorad, Hercules, Ca, USA, at 230 V 1mF) or the Neon™ transfection system (Invitrogen, Karlsruhe, Germany, using one pulse at 1100 V for 30 ms). Cells were patch-clamped in whole cell recording modus using an EPC-9 patch-clamp amplifier controlled by the PULSE software (Heka Elektronik, Lambrecht, Germany). The extracellular solution contained (in mM): 146 NaCl, 2.4 KCl, 10 HEPES, 1.2 $MgCl_2$, 2.5 $CaCl_2$, 10 glucose and 10 $NaHCO_3$ (pH 7.4, 310 mOsm). Secretion was induced through either depolarization trains[62] or perfusion of the cells with 6 µM $Ca^{2+}$ containing solution via the patch-clamp pipette[14]. The intracellular solution contained (in mM) either (experiment from ref. [14]) 160 Cs-aspartic acid, 10 HEPES, 1 $MgCl_2$, 2 Mg-ATP, 0.3 $Na_2$-GTP (pH 7.2, 300 mOsm) or

(experiment from ref. 62) 110 Cs-glutamate, 10 HEPES, 2 Mg-ATP, 0.3 Na2-GTP, 5 CaCl$_2$, 9 HEDTA (pH 7.2, 300 mOsm). All products for the solutions were from Sigma-Aldrich/Merck. The acquisition rate was 10 Hz and the exposure time was 100 ms. The camera was either a Micromax 512BFT camera (Princeton Instruments Inc., Trenton, NJ, USA) with 100 × /1.45 NA Plan Apochromat Olympus objective[62], or a QuantEM 512SC camera (Photometrics, Tucson, AZ, USA) with an 100 × /1.45 NA Fluar (Zeiss) objective[14], giving a final pixel size of 130 or 160 nm$^2$ respectively.

## INS-1 cells

**Culture and transfection.** Rat insulinoma cells[64] (INS-1 cells, clone 832/13 provided by Hendrik Mulder, Lund University) were maintained in RPMI 1640 (Invitrogen, Cat#21870076) containing 10 mM glucose and supplemented with 10% fetal bovine serum(Sigma-Aldrich, Cat# F7524), streptomycin (100 µg/ml) and penicillin (100 µg/ml, Biowest, Cat# L0022), Na-pyruvate (1 mM, Gibco, Cat# 11360-070) L-glutamine (2 mM, Biowest, Cat# X0550), HEPES (10 mM, Gibco, Cat# 15630-080) and 2-mercaptoethanol (50 µM, Gibco, Cat# 31350-010). The cells were plated on polylysine-coated coverslips (Sigma-Aldrich, Cat# P5899 and Marienfeld, Cat# 112620), transfected using lipofectamine 2000 (Invitrogen, Cat#11668-019) with a ratio of 0.1 µg DNA:1 µl lipofectamin, and imaged 24-42 h later.

**Acquisition conditions.** The bath solution contained (in mM) 138 NaCl, 5.6 KCl, 1.2 MgCl$_2$, 2.6 CaCl$_2$, 10 D-glucose, 0.2 diazoxide (Sigma, Cat# D9035), 0.2 forskolin (Merk, Cat# 93049), and 10 HEPES (Sigma, Cat#H4034-1KG), pH 7.4 adjusted with NaOH. Individual cells were stimulated by computer-controlled air pressure ejection of a solution containing elevated K$^+$ (75 mM replacing Na$^+$) through a pulled glass pipette (similar to patch clamp electrode, Hilgenber, Cat# 1003027) that was placed near the recorded cell. The bath solution temperature was kept at 35 °C using a FCS13-A electronic heater (Shinho, Cat# FCS11E7 2002.07).

**Imaging.** INS1 cells (clone 832/13) that transiently expressed NPY-mGFP, NPY-mNeonGreen or NPY-mCherry were imaged using a custom-built lens-type total internal reflection (TIRF) microscopes based on AxioObserver D1 microscope with an x100/1.46 objective (Carl Zeiss, Cat# 420792-9800-720). Excitation was from a diode laser module at 473 nm, or a diode pumped laser at 561 nm, respectively (Cobolt, Göteborg, Sweden, Cat# 0473-06-01-0300-100 & Cat# 0561-06-91-0100-100), controlled by an acoustic-optical tunable filter (AOTF, AA-Opto, France, Cat#AOTFnC-400 650-TN). Light passed through a dichroic Di01-R488/561 (Semrock), and emission light was separated onto the two halves of a sCMOS camera (Prime 95B, Photometrics, Tucson, AZ, USA, Cat# 01-PRIME-95B-R-M-16-C) using an image splitter (Dual view, Photometrics) with a cutoff at 565 nm (565dcxr, Chroma) and emission filters (FF01-523/610, Semrock; and ET525/50 m and 600EFLP, both from Chroma). Scaling was 110 nm per pixel (sCMOS camera). The acquisition rate for NPY-mNeonGreen was 50 Hz and 10 Hz for NPY-mCherry. NPY-eGFP expressing INS1 cells were imaged using a TIRF microscope that was based on an AxioOb-server Z1 (Zeiss) with a diode pumped laser at 491 nm (Cobolt, Stockholm, Sweden, Cat# DC-4915615050-300) that passed through a cleanup filter and dicroic filter set (zet405/488/561/640x, Chroma). Imaging was done with a 16-bit EMCCD camera (QuantEM 512SC, Roper) with a final scale of 160 nm per pixel. The acquisition rate was 10 Hz. Image acquisition was conducted with MetaMorph (V7.8.0.0, Molecuar Devices).

## Human CD8 + T lymphocytes

**Cells.** Human CD8 + T cell clones were used as cellular model. Human T cell clones were isolated and maintained as previously described[65].

Briefly, cells were cultured in RPMI 1640 medium GlutaMAX (Gibco, Cat# 61870036) supplemented with 5% heat inactivated human AB serum (Institut de Biotechnologies Jacques Boy, Cat# 201021334), 50 µM 2-mercaptoethanol (Gibco, Cat# 31350010), 10 mM HEPES (Gibco, Cat# 15630122), 1× MEM-Non-Essential Amino Acids (MEM-NEAA) (Gibco, Cat# 11140035), 1× sodium pyruvate (Sigma-Aldrich, Cat# S8636), ciprofloxacin (10 µg/ml, Sigma-Aldrich Cat# 17850), human recombinant interleukin-2 (rIL-2; 100 IU/ml, Miltenyi Biotec Cat# 130-097-748), and human rIL-15 (50 ng/ml, Miltenyi Biotec, Cat# 130-095-766). Blood samples were collected and processed following standard ethical procedures after obtaining written informed consent from each donor and approval by the French Ministry of the Research as described (Cortacero et al. 2023, authorization no. DC-2021-4673).

**Acquisition conditions.** Human CTLs were stained for 30 min with Lysotracker red (DND-99) dye (2 µM, Invitrogen Cat# L7528) at 37 °C/5% CO$_2$. The cells were washed 3 times with RPMI 1640 medium (1X) w/o pH Red (Gibco, Cat# 11835063) supplemented with 10 mM GlutaMAX (Gibco, Cat# 35050061) and 10 mM of HEPES (Gibco, Cat# 15630122). To induce immunological synapse formation followed by lytic granule exocytosis µ-Slide 15 Well 3D glass bottom slides (Ibidi, Biovalley Cat# 81507) were coated with poly-D-lysine (1:10, Sigma-Aldrich Cat# P6407), human monoclonal anti-CD3 antibody (TR66) (5 µg/mL or 10 µg/mL, Enzo Life Sciences Cat# ALX-804-822) and recombinant human ICAM-1/CD54 Fc Chimera Protein (5 µg/mL or 10 µg/mL, R&D Systems Cat# 720-IC) at 4 °C overnight. The chambered slides were washed 3 times with PBS 1X (Sigma-Aldrich Cat# D8537) and mounted on a heated stage within a temperature-controlled chamber maintained at 37 °C and constant 5% CO$_2$. For each recording, $3 \times 10^4$ to $5 \times 10^4$ cells were seeded on the chambered slides. During acquisition, the cells were in RPMI 1640 medium (1X) w/o pH Red supplemented with 10 mM GlutaMAX, 10 m HEPES and 5% Fetal Bovine Serum (FBS, Gibco, Cat# A5256701).

**Imaging.** The TIRFM set up acquisition was based on an Eclipse Ti2-E inverted microscope (Nikon Instruments) equipped with a 100 × /1.45 NA Plan Apochromat LBDA objective (Nikon Instruments) and an iLAS 2 illumination control system (Roper Scientific SAS). A diode laser at 561 nm (150 mW) (Coherent) band-passed using a ZET405/488/561/647x filter (Chroma Technology) was used for excitation. The emissions were separated using a ZT405/488/561/647rpc-UF1 dichroic mirror (Chroma Technology) and optically filtered using ZET405/488/561/647 m filter (Chroma Technology). Images were recorded on a Prime 95B Scientific CMOS Camera (Teledyne Photometrics, Tucson, AZ, USA). The final pixel size was 110 nm. Image acquisition was controlled using MetaMorph Software (Version 7.10.5.476, Molecular Devices) and Modular V2.0 GATACA software. The acquisition frequency was 9 Hz for a duration of 20 to 30 min.

## Dopaminergic neurons

Data showing dopaminergic neuron exocytosis monitored by Andro-meDA nanosensor paint technology were from Elizarova et al.[39]. Briefly, ventral midbrain neurons were dissected from postnatal day 0 C57BL/6 mice and enzymatically dissociated using papain (Worthington, Cat# 9001-73-4). Cells were plated on glass coverslips pre-coated with poly-L-lysine (Sigma-Aldrich, Cat# P4707-50ML) and maintained in Neurobasal-A medium (Gibco, Cat# 11540366) supplemented with B-27 (Gibco, Cat# 17504-044), GlutaMAX (Gibco, Cat# 35050-038), and penicillin-streptomycin (Gibco, Cat# 15140-130). Neurons were cultured at 37 °C in a humidified 5% CO$_2$ atmosphere and imaged between DIV 21 and DIV 42. The imaging setup included a 100× oil-immersion objective (UPLSAPO100XS, Olympus) and a Xenics Cheetah-640-TE1 InGaAs camera (Xenics), yielding a final pixel size of 150 nm. Imaging was performed at 15 Hz.

## Cardiomyocytes from mouse

Data showing Fluo4 measured calcium sparks in mouse cardiomyocytes were from Tian and Lipp[44]. Mouse ventricular myocytes were isolated as previously described with full details[66]. All the procedures concerning animal handling conformed to the guidelines from Directive 2010/63/EU of European Parliament. After isolation, the cardiomyocytes were rinsed in equilibrium solution (in mM, 140 KCl, 0.75 $MgCl_2$, 0.2 EGTA, 10 HEPES, pH 7.20; from Sigma-Aldrich/Merck) for 2 ~3 min to let the cells settle. The cells were then rinsed with saponin (ChemCruz, Cat# sc-280079, 15 pg/ml dissolved in equilibrium solution) for 40 s. After that the solution was completely removed and exchanged with artificial internal solution (in mM, 100 K-Aspartate, 15 KCl, 5 $KH_2PO_4$, 0.5 EGTA, 10 HEPES, 10 phosphocreatine (CalBioChem, Cat# 2380), 8% ~40,000 MW dextran (Sigma, Cat# 31389), 5 MgATP, 5 U/mL creatine phosphokinase (CalBioChem, Cat# 2380), 10 µM Fluo-4 (Invitrogen, Cat# F14200)). pH was set to 7.2 and the free $Ca^{2+}$ was calibrated to 100 nM. For more details, please refer to[67]. The data was recorded with 2D-array scanning confocal microscope (Infinity4, Visitech, Sunderland, UK) equipped with a NIKON 60x oil immersion objective (NA = 1.40) and a sCMOS Flash4 V2 camera from Hamamatsu (Hamamatsu Photonics Deutschland GmbH, Herrsching am Ammersee, Germany). In the camera settings, two by two binning (0.217 × 0.217 µm/pixel) was used when the imaging was done with the imaging software VolCell (v8.03.0.3, Visitech, Sunderland, UK). Suitable area of the camera chip was selected such that a final speed of 124 frames per second was achieved.

## Statistical analysis and used programs and algorithms

All statistical analyses were performed with SigmaPlot (V14.5.0.101, Systat Software, Inc.). *P*-values were calculated with two-tailed statistical tests and 95% confidence intervals. ANOVA, ANOVA on ranks and Student's *t*-test were used as required. Data analysis and processing was performed with MATLAB (Mathworks 2024b) and Excel 2021 (Microsoft (V2108)).

IVEA was developed in Java 1.8.0_322 using Eclipse and the following Java libraries: ij (V1.54c), opencsv, bio-formats_plugins, loci_plugins, deeplearning4j core v1.0.0-M1.1, Google TensorFlow v1.15.0, libtensorflow_jni v1.15.0.

IVEA for training was done in Python v3.8.15 language using with the following libraries: Google TensorFlow v2.9.1 or v2.10, Keras, Numpy, Scikit-image, Tkinter, shutil, pandas, h5py, and read_roi.

IVEA training platform was coded with visual studio code V1.100.2

We used deep learning long-short term memory network, vision transformer network, convolution neural network, k-means clustering, iterative thresholding, Gaussian non-maximum suppression and multilayer intensity correction algorithm for software development.

Imaging data were analyzed by the human expert with Fiji V1.54p. The results were compared to ExoJ (V1.09), pHusion and SynActJ V0.3 software.

Figures were prepared with CorelDraw V23.5.0.506 and Adobe Illustrator V29.5.1.

## Ethics statement

Mice were treated according to the regulations of the local authorities, the state of Saarland (Landesamt für Verbraucherschutz) under the license AZ.: 2.4.1.1 or the Niedersächsisches Landesamt für Verbraucherschutz und Lebensmittelsicherheit (LAVES, permit numbers 33.19-42502-04-19/3254, 33.19.42502-04-15/1817 and 33.19-42502-04-18/2756). Animals were housed according to European Union Directive 63/2010/EU and ETS 123 at 21 + /- 1 °C, 55% relative humidity, under a 12 h/12 h light/dark cycle, and received food and tap water ad libitum. Human blood samples were collected and processed following standard ethical procedures after obtaining written informed consent from each donor and approval by the French Ministry of the Research (authorization no. DC-2021-4673).

## Reporting summary

Further information on research design is available in the Nature Portfolio Reporting Summary linked to this article.

## Data availability

All original datasets used in this study, including the Source Data file, the labeled training data, and demonstration videos, have been deposited on *Zenodo*: https://doi.org/10.5281/zenodo.13153017. Data will be made available without restriction upon request via the *Zenodo* web interface. However, the underlying data for Fig. 3 and Supplementary Data 4 will be provided without restriction only to programmers who require access to a large amount of data for testing and training new models. Source data are provided with this paper.

## Code availability

The code used to develop the model, perform the analyses, and generate the results in this study is publicly available under the GPL v3.0 license at GitHub: https://github.com/AbedChouaib/IVEA. The version of the code associated with this manuscript has been archived on *Zenodo* at https://doi.org/10.5281/zenodo.15498139[68].

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

## Acknowledgements

This work was supported by grants from the Deutsche Forschungsgemeinschaft (SFB 894 to U.B.), the European Commission (ERC-2021-SyG_951329 to the Department of Cellular Neurophysiology, Saarland University and to S.V., Cancer Research Center of Toulouse, INSERM UMR1037) and grants from University of Saarland (HOMFORexzellent2020 and NanoBioMed Young Investigator grant 2020 to H.-F.C.). A.H.S. was funded by the European Research Council (ERC) under the European Union's Horizon 2020 research and innovation program (grant agreement No 951275). C.P. was supported by the Deutsche Forschungsgemeinschaft (DFG, German Research Foundation) under Germany's Excellence Strategy - EXC 2067/1-390729940. Q.T. was funded by the Medical Faculty of Saarland University (HOMFORexcellent). P.L. was supported by DFG (SFB894/A19). S.B. was supported by the Swedish Science Council, Diabetes Wellness Network Sweden, the Swedish Diabetes Society, the NovoNordisk Foundation, and Excellence of Diabetes Research in Sweden (EXODIAB). We thank Hindrik Mulder, Lund University for providing the INS-1 cells. We thank Marcel Lauterbach and Silvio O. Rizzoli for their constructive comments on the initial version of the manuscript. We thank Margarete Klose, Anja Bergsträßer, Nicole Rothgerber for excellent technical assistance.

## Author contributions

A.H.S. and U.B. conceived and supervised the project. A.A.C. developed the methodology, investigated and analyzed the data. C.P. provided critical input on the developed artificial intelligence strategies. A.A.C., U.B, H.-F.C., O.M.K, N.A., L.D., S.Ec., Q.T. were the human experts. H.-F.C., O.M.K., N.A., Q.T. S.Ec., L.D., S.El., J.A.D., Q.T., A.H.S. and U.B. acquired the original biological data. U.B. acquired and managed the project funding. Additional funding was acquired by S.V., S.B., Q.T., P.L. and C.P. A.A.C, A.H.S and U.B. wrote the initial draft of the manuscript which was edited by E.F.F. and H.-F.C. All authors revised the manuscript.

## Funding

## Competing interests

The authors declare no competing interests.
