## [Transparent Peer Review file · Nature Communications]

Highly adaptable deep-learning platform for automated detection and analysis of vesicle exocytosis

Corresponding Author: Dr Ute Becherer

Version 0:

Reviewer comments:

Reviewer #1

(Remarks to the Author)

The manuscript by Chouaib et al. describes a program called IVEA, distributed as an open source plugin for the widely used image analysis tool ImageJ, to detect and quantify exocytic events recorded by fluorescence imaging of cultured cells. A powerful method to detect exocytosis of various organelles (secretory vesicles, recycling endosomes, lysosomes, multi-vesicular bodies, synaptic vesicles) has been to image with fluorescence microscopy markers of these organelles tagged with pH sensitive (in the present manuscript granzymeB-pHuji, synaptophysin-pHluorin) or pH insensitive tags (granzymeB-ttdTomato, NPY-mCherry) as well as sensors for secreted cargo (here the dopamine sensor AndromeDA). The program enables the automated detection of these events, based on neural network models built by annotation of a large number of events by human experts (HEs). This program differs from other existing programs detecting and characterizing exocytosis events (cited in the Introduction) by the fact that, in routine use, it promises a fully automated analysis.

I have read the manuscript in full, installed and tried the IVEA plugin on data collected in my laboratory. It is a cell transfected with CD63-pHluorin and imaged with TIRF microscopy for 1 min at 10 Hz frame rate, which would resemble the CTL transfected with GranzymeB-pHuji shown in Figure 3a. In this cell, I could extract, with homemade Matlab based software, 63 events. Moreover, I have used the recently published ExoJ plugin (Liu et al. J Cell Science 2024 10.1242/jcs.261938) with relative ease and found almost the same number of events. With the IVEA plugin, using the model1 and default parameters, I could detect only 5 events. Using the model2, I could detect 17 events, but it is unclear why these and not others (as bright and distinct as the detected ones) were not detected. It was difficult to tune parameters to optimize performance, because one can only observe the end result and none of the intermediate steps. Presumably, the neural network models simply named "model1, model2", these are the EViT and LSTM models described in the manuscript. The authors need to explicitly name these models and give the parameters used to train them more precisely than what is provided in the "Neural network training" section of the Materials. Moreover, the indications for "training new data type" are too general to be useable to build new neural networks without assistance from the authors.

I understand that for the authors, the IVEA plugin is powerful in detecting exocytosis events in their conditions (cell type, marker, imaging modality). However, for applicability for the community interested to use this tool, the description of the tools, how they are built, are not clear enough to be useable in the present stage. The user interface and user manual also lack many details to make it useable.

Overall, I found that the article difficult to follow with a lot of jargon which obscures the understanding. Mixing three methods for detecting fairly different types of imaging is confusing. It will be much clearer to describe the first type of recordings (non-fixed FBE) first, then add the particularities of the other types of recordings. Finally, the figures do not correctly describe the expected types of events and their treatment by the algorithms. For example, Figure 1 shows "image patches": are they extracts of the parent video, consecutive frames? What are the "image patches" leading to the event patterns depicted in the 13 regions? What is the rationale of choosing 13 regions?

For non-fixed FBE analysis, the authors present performance of their algorithm on 4 types of recordings with pH sensitive and pH insensitive probes (Figure 3). Presumably, the neural network models used for these analyses are different. This should be clearly explained. Moreover, the authors conclude that "IVEA detected 246 additional events from those identified by the HE [initially 673]". The authors should describe if the events detected by IVEA are any different than the ones initially detected by human experts.

Regarding style, half of the introduction, starting with "The event detection method..." describes the IVEA software. It should be in the Results section, with justification of the parameters used.

In conclusion, although a truly automated and efficient method is a major advance in the field of live cell fluorescence imaging, in particular for imaging exocytosis, the manuscript needs major rewriting before I can recommend it for publication.

(Remarks on code availability)

I could install the plugin, run it on my computer with data acquired in my lab. I could get some detected events but I have issues (see main review)

Reviewer #2

(Remarks to the Author)

Using machine learning, the authors developed a new method called "Intelligent Vesicle Exocytosis Analysis (IVEA)" to detect and analyze vesicle exocytosis events in living cells. IVEA integrates into the Fiji platform for open-source accessibility. The authors demonstrated superior capabilities of IVEA to detect vesicle exocytosis faster and more accurately in several images in different types of cells. Thus, IVEA will significantly help with imaging analyses of vesicle exocytosis, which will contribute to the scientific community. This manuscript could be worth publishing in Nature Communications. However, several issues should be addressed before it is accepted.

One of major issues is whether IVEA can be applied to fluorescence images other than a pH-sensitive GFP to detect vesicle exocytosis events. As the authors may know, there are several other exocytosis indicators including FM, quantum dots and CypHer5E (please see the following references: (Murthy 1999, Hua, Sinha et al. 2011, Yu, Zhang et al. 2016, Jang, Kim et al. 2021, Park, Jung et al. 2022)). The authors should test whether IVEA can be used to detect vesicle exocytosis in images based on other exocytosis markers in order to support the authors' claim – "IVEA has the potential to revolutionize the analysis of fluorescence fluctuations that occur during exocytosis, regardless of microscope type or fluorophore dye characteristics, with unprecedented accuracy and efficiency."

Another issue is that the authors used similar imaging conditions for vesicle exocytosis for IVEA. Many laboratories used 1Hz imaging frequency to image vesicle exocytosis in living cells instead of 10Hz imaging. It is possible that IVEA may not properly detect vesicle exocytosis at different experimental conditions. Thus, the authors should test whether IVEA is capable of detecting vesicle exocytosis in images obtained at different experimental conditions.

The other issue is that IVEA may not detect non-burst exocytosis events shown in hippocampal neurons. Not detecting non-burst exocytosis events in neurons could restrict the application of IVEA in neuroscience. I strongly encourage that the authors should further improve IVEA to detect non-burst exocytosis events in neurons for neuroscientists.

Here are some minor issues.

1. What stimulation protocols were used to cause fluorescent burst events (FBE)?
2. The text from line 72 to 166 should be placed in Results instead of Introduction.
3. The authors should describe how the authors calculate recall, precision and F1 score in details.
4. What is the size of the pixel in line 190?
5. Figure 2b, 4a and 5c were not mentioned in the main text.
6. Font of "a, b, c, d" in main figures are too small.
7. Figure 5 used "(a), (b), ..." instead of "a, b, ..." in other figures. The authors should show the coherence in figures.
8. In line 519, "is capable to" "is capable of"

References

- Hua, Y., R. Sinha, C. S. Thiel, R. Schmidt, J. Hüve, H. Martens, S. W. Hell, A. Egnér and J. Klingauf (2011). "A readily retrievable pool of synaptic vesicles." *Nat Neurosci* 14(7): 833-839.
- Jang, Y., S. R. Kim and S. H. Lee (2021). "Methods of measuring presynaptic function with fluorescence probes." *Appl Microsc* 51(1): 2.
- Murthy, V. N. (1999). "Optical detection of synaptic vesicle exocytosis and endocytosis." *Curr Opin Neurobiol* 9(3): 314-320.
- Park, C., S. Jung and H. Park (2022). "Single vesicle tracking for studying synaptic vesicle dynamics in small central synapses." *Curr Opin Neurobiol* 76: 102596.
- Yu, C., M. Zhang, X. Qin, X. Yang and H. Park (2016). "Real-time imaging of single synaptic vesicles in live neurons." *Frontiers in Biology* 11(2): 109-118.

(Remarks on code availability)

We tested IVEA and it works very well.

Reviewer #3

(Remarks to the Author)

With pleasure I was reading a manuscript about the developed "Intelligent Vesicle Exocytosis Analysis" (IVEA) platform for analysing vesicle fusion events in fluorescence microscopy images. Obviously, automation for these kinds of tasks is very useful, not just because they spare time, but also because it eliminates inter-observer variability. The fact that the authors developed their tool as Fiji plugin is exciting to note and will ensure that the community can adopt the tool.

General feedback:

* As the tool was developed open-source for the Fiji platform, the authors could highlight early in the manuscript that it is available open source as common in the Fiji community.

* The manuscript introduces multiple deep-learning based techniques, special new developed architectures and models. Hardly anything is written about how these were trained (e.g. line 242) and validated and which data was used for this. How many human experts were involved in producing the training data? What was their qualification to produce high-quality training data? Will the training data be available open access so that others can improve the state-of-the-art in the coming years?

* Ablation studies: In computer science the concept of ablation studies is established to demonstrate that new methods are not overengineered or overfitting. If one develops a new algorithm / workflow that consists of many layers, paradigms and modules, single individual modules are deactivated in a test-run of the algorithm to see this step contributes to the overall performance of the tools. The presented tools are quite complex and I'm wondering if a combination of preprocessing + ViT + LSFTM + machine learning is really necessary to solve the scientific question. An ablation study might help.

More detailed feedback:

Abstract:

* Line 23 "human capital" - maybe "human resources" or "work effort" might sound better?

Introduction

* Line 43/44: "ion flux" depending on the target audience of the article, a sentence explaining what this is might be helpful. I'm a computer scientist and only have a vague idea.

* Line 104: "which involves machine learning". After the deep-learning approaches in the lines before were more precise (using ViT and LSTM), I think it would be good to write here which algorithm was used as "machine learning".

* Line 105/106: Apologies for my ignorance, but I think an explanation of "nanosensor-based technologies" would be great.

* Line 108: "forward and backward subtraction" Could you explain what this is? Maybe an equation or a figure helps explaining the process. I suspect this is about image subtraction over time, but I'm not sure.

* Line 115: "we select for each image regions": Is this a manual process? If not, what algorithm is used for this?

* Line 118: "a ViT network, which classifies them as exocytosis or not". How was the training of this network done? How many images / annotations were necessary to train it?

* Line 119: "developed for our ViT architecture (21)". The citation here suggests that the cited work was done by the same authors as the manuscript I'm reading. But I see no overlap between the author lists. May rephrasing this can clarify.

Results

* Lines 168-180: This section reads like a Materials&Methods section and not Results.

* Line 192: Same here. I would expect equations and explanations about what was done in a Methods section.

* Line 258: The explanation of "event's cloud spread" also seems a method, not a result.

* Line 291-311: This section is hard to read and follow. Would it be possible to put this data in a table and underline important results? E.g. the best method investigating a certain question.

* Line 318: The comparison of the two developed method seems nice. I'm just curious if a comparison with methods developed by others would make sense?

* Line 324ff. Here comes a comparison with ExoJ but the context to the methods before is a bit vague. A short clear statement of all compared methods and their advantages and disadvantages would be great.

* Line 343: I presume, the CSV file also contains the coordinates of the center of mass?

I must admit that I couldn't review the entire article due to time constraints. I nevertheless hope my feedback is helpful.

Sincerely,
Robert Haase
robert.haase@uni-leipzig.de

(Remarks on code availability)

I did not have the time to test the tool, the website offers tutorials and example data, which is a very good sign.

Version 1:

Reviewer comments:

Reviewer #1

(Remarks to the Author)

I have read the revised manuscript together with the answer of the authors. I acknowledge the effort made by the authors to improve readability and better use of the plugin. I have no further comment on the manuscript, except another round of editing for numerous typos.

However, I find that the output given by IVEA, merely a list of event names, is minimal. The authors should consider adding more features for easy analysis of the data, such as the ones provided by the alternate plugin for analysis of exocytosis events ExoJ (local fluorescent measures, basic features of each exocytosis event, etc...).

(Remarks on code availability)

I have downloaded the new version of the IVEA plugin and used it to analyse a cell recorded in my lab with the GranuVision3 model. It ran successfully on my computer

Reviewer #2

(Remarks to the Author)

The revised manuscript improved. The authors addressed several concerns raised by the reviewers. However, some issues should be addressed before it is accepted. Here are some issues to be addressed before acceptance.

It is disappointing that the authors did not apply IVEA to other exocytosis indicators including FM, quantum dots and CypHer5E because these exocytosis markers are widely used in many labs. I thought that this could be done by receiving raw images from research groups which have used FM, quantum dots and CypHer5E for their exocytosis assays. If the authors do not want to test whether IVEA is capable to detect exocytosis events in images from these FM, quantum dots and CypHer5E, the authors should mention the usage of these indicators to detect exocytosis and should write that IVEA has not been tested for these exocytosis indicators in Discussion of the revised manuscript clearly. Furthermore, the authors also should discuss the possibility of the application of IVEA to these endocytosis markers. Then, the readers will have better understanding of the application of IVEA.

It is discouraging that the authors did not include the new method to detect non-burst exocytosis events in neurons. The authors should discuss it in the revised manuscript. The discussion does not need to be lengthy. It will help the readers a lot.

The Editor asked me to comment on the author's response to Reviewer 3's comments.

The authors seemed to address Reviewer 3's comments appropriately. There is no further comment.

(Remarks on code availability)

It works well.

Point-by-point reply to the reviewers

Reviewer comments are in *italics* and our responses are in regular font. Changes in the manuscript are highlighted in light blue color. Page and line number are given on the document **without** “track change”.

Reviewer #1 (Remarks to the Author):

The manuscript by Chouaib et al. describes a program called IVEA, distributed as an open source plugin for the widely used image analysis tool ImageJ, to detect and quantify exocytic events recorded by fluorescence imaging of cultured cells. A powerful method to detect exocytosis of various organelles (secretory vesicles, recycling endosomes, lysosomes, multi-vesicular bodies, synaptic vesicles) has been to image with fluorescence microscopy markers of these organelles tagged with pH sensitive (in the present manuscript granzymeB-pHuji, synaptophysin-pHluorin) or pH insensitive tags (granzymeB-tdTomato, NPY-mCherry) as well as sensors for secreted cargo (here the dopamine sensor AndromeDA). The program enables the automated detection of these events, based on neural network models built by annotation of a large number of events by human experts (HEs). This program differs from other existing programs detecting and characterizing exocytosis events (cited in the Introduction) by the fact that, in routine use, it promises a fully automated analysis.

We would like to thank the Reviewer for the encouraging evaluation and the comments. We would also like to thank them for the helpful feedback which we now address in detail, significantly improving the quality of our work.

I have read the manuscript in full, installed and tried the IVEA plugin on data collected in my laboratory. It is a cell transfected with CD63-pHluorin and imaged with TIRF microscopy for 1 min at 10 Hz frame rate, which would resemble the CTL transfected with GranzymeB-pHuji shown in Figure 3a. In this cell, I could extract, with homemade Matlab based software, 63 events. Moreover, I have used the recently published ExoJ plugin (Liu et al. J Cell Science 2024 10.1242/jcs.261938) with relative ease and found almost the same number of events. With the IVEA plugin, using the model1 and default parameters, I could detect only 5 events. Using the model2, I could detect 17 events, but it is unclear why these and not others (as bright and distinct as the detected ones) were not detected. It was difficult to tune parameters to optimize performance, because one can only observe the end result and none of the intermediate steps. Presumably, the neural network models simply named “model1, model2”, these are the EViT and LSTM models described in the manuscript. The authors need to explicitly name these models and give the parameters used to train them more precisely than what is provided in the “Neural network training” section of the Materials. Moreover, the indications for “training new data type” are too general to be useable to build new neural networks without assistance from the authors. I understand that for the authors, the IVEA plugin is powerful in detecting exocytosis events in their conditions (cell type, marker, imaging modality). However, for applicability for the community interested to use this tool, the description of the tools, how they are built, are not clear enough to be useable in the present stage. The user interface and user manual also lack many details to make it useable.

Thank you for your detailed feedback and for taking the time to test our IVEA plugin with your data. We have carefully considered all your comments and made four improvements to address all the issues you raised:

1. updating the interface and the manual to enhance the user's experience

We understand the confusion of the reviewer and addressed it by describing each analysis paradigm separately in the manuscript. In addition, the GUI of the software has been modified by dividing each analysis paradigm into its own section in the “IVEA” menu, thus creating a modular architecture. Moreover, the GUI now provides tooltip messages and ImageJ log messages, which are designed to facilitate enhanced insights on the parameter modification by users. These modifications aim to enhance the accessibility of the IVEA plugin to a wider community. Please see modified Supplementary Figure 2 and 3 for an overview. We also updated IVEA's manual to make it more understandable and to include all the changes we made in IVEA see <https://github.com/AbedChouaib/IVEA>.

2. Solving the detection issues

The type of movies that you tried to analyze with our program display exocytosis events that correspond to “non-fixed FBE” now referred to as “random burst events”. Indeed, in those movies the vesicles were labeled for example by CD63-pHluorin, they move throughout the cell and get exocytosed at any random spot on the plasma membrane. Therefore they are best analyzed by the eViT analysis paradigm.

The exocytosis events in these movies present a very specific fluorescent intensity profile that was new to the old model (old model 1, now named **GranuVision2**). The utilization of pHluorin as a fluorescent marker, in conjunction with low intravesicular pH, rendered the vesicles invisible prior to exocytosis. This confounded the older models, which were trained on vesicles that were visible before exocytosis. The exocytosis events were marked by a pronounced increase in fluorescence intensity, often without accompanying clouds, which may have been a challenge for the older models. Furthermore, CD63 serves not only as a marker for the vesicle membrane but also for the released exosome. Upon release, these exosomes tend to remain at the site of release, generating a prolonged, elevated fluorescent signal, which further complicates the interpretation of the older models. This slow decrease of fluorescence was associated with vesicle undocking.

To address these shortcomings of our old models, we have retrained our eViT on new data and generated a model that recognizes events specific to tetraspanin-XFP, such as CD63-pHuji or CD63-SEP, overexpression namely the model **GranuVision3**. We kept the old model named **GranuVision2**, which is used in the manuscript (Fig 3a, b, d, and e) and performed well for most needs. As can be seen in Fig 3c and Supplementary Table 3, with the new model we achieved a recall of $97.84 \pm 1.39\%$, a precision of $98.58 \pm 1.30\%$, and an F1 score of $98.16 \pm 1.07\%$ for CD63-XFP labeled events, outperforming the ExoJ plugin (recall: $96.23 \pm 2.71\%$, precision: $87.48 \pm 9.97\%$, F1: $88.92 \pm 7.28\%$). For long-lasting and slow events, we down sampled the movie frame rate from 10 Hz by a factor of 5, to 2 Hz, which allowed the spatiotemporal features to fit in the 28 frames of the **GranuVision3** model. For the user we have added in the GUI a video downsampling function based on temporal max pooling, i.e. a moving maximum intensity projection. This enables frame reduction of videos without losing some important features such as the clouds. Furthermore, this function works only in the detection method without impacting the original video or the final ROI/measurement temporal information.

3. Renamed models to increase clarity

We chose the model naming to convey the information on the type of network that is used “LSTM” vs “Vision” for the eViT and the events it can detect, “Granu” for the exocytosis of individual granules (Lytic granules or large dense core granules) and “Neuro” for release at neuronal synapses. The default model for random burst events is **GranuVision3**, as it is the most versatile

and performs well across different conditions. For stationary burst events (i.e. synaptic transmission) the default model is “NeuroLSTM”.

We now provide a Table with the model description (Supplementary Table 1) and detailed descriptions of the eViT models architecture and naming in Methods section “Encoder-ViT network architecture” page 21 Line 682-715. This information should help users better understand how the models were built and how they function.

For convenience we are also adding the text here:

Encoder-ViT network architecture

Our eViT network consists of two components: a convolutional neural network (CNN) as the encoder for feature extraction from image patches and a vision transformer (ViT) for classification. The encoder is a shared CNN. It comprises seven layers, including a 2D spatial convolution layer that is followed by a sequence of 3D convolution layers and 3D max pooling operations (Supplementary Fig. 14a,b).

We have 2 pretrained models available with IVEA, namely GranuVision2 and GranuVision3. The model’s encoder input layer accepts time series of 26 or 28 image patches for the GranuVision2, or the GranuVision3, respectively. The width and height of each patch is $32 \times 32 \times 1$, with the last dimension corresponding to the number of channels, such as $X \in \mathbb{R}^{(t \times w \times h \times c)}$. If the dimension of the image patches changed, we use bilinear interpolation to resize the images to 32×32 . The initial stage of the shared CNN involves the application of a 3D convolution layer with 16 filters and ReLU activation, which is followed by a 3D max pooling layer. Subsequently, another 3D convolution layer with 32 filters with ReLU activation, followed by another 3D max pooling operation. Finally, a 3D convolution layer with 64 filters and ReLU activation, followed by a concluding 3D max pooling layer. Each batch of the encoded image patches is passed through a Flatten layer and a Dense layer of size 64, then positional embedding is added to the data before passing them into a transformer block and subsequently into an MLP. The transformer block consists of multi-head self-attention and an MLP, each followed by residual connection (Fig. 1b). The transformer block operates on an input sequence of length equal to the time series dimension, with a key dimension representing the output size of the previous Dense layer. The MLP inside the transformer block consists of two Dense layers, where the first layer has a dimension twice the key dimension, and the second layer projects it back to the key dimension. The last MLP is comprised of two Dense layers, the first layer has GeLU non-linearity, while the second has a SoftMax activation for classifying 10 (2 fusion + 8 artifacts) or 11 (fusion + 8 artifacts) classes for the GranuVision2, or the GranuVision3, respectively.

4. Clarification on how the training was done and the model evaluated

We now given more details on the training of the model pages 22-24 lines 731-802 in the methods section “*Neural network training and evaluation*”.

5. New eViT training

The Python IVEA training GUI has been updated to offer a more streamlined experience, featuring options for refining existing models or developing new ones. Additionally, in IVEA’s Python main directory, we provide JSON files containing all the default parameters necessary for training, as well as the dataset file Hierarchical Data Format (h5) used for model training. We have also integrated plugins for labeling ROI files and performing profile intensity measurements in ImageJ. These plugins are accessible through the Plugins-IVEA menu in ImageJ, making it easier to label new data and verify exocytosis over specified time intervals. This integration is referred to as the IVEA ImageJ package and it is described page 24 lines 803-817. This platform is available on our internet site (<https://cloud.hiz-saarland.de/s/eEaF4A8eWpr88Qf>, password IVEA@V2_2025) and will be available on GitHub upon acceptance of the manuscript. This now indicated page 23,

lien 765-766 as follows: “For generating a new model, training files and tool are available on our GitHub page for IVEA (see data available section)”.

Furthermore, we enhanced the new training capability of IVEA by offering the option to refine existing model through transfer learning. This allows the user to quickly adjust existing model to their specific need. We successfully tested the refinement procedure on GranuVision3 for recognizing calcium sparks in cardiomyocytes. The result is presented Supplementary Fig. 8 and in the result page 10 lines 324 – 330 as follows: “By learning complex spatiotemporal patterns via new training or by adapting pretrained models via a refinement process, eViT adapts to diverse fluorescence signals (Supplementary Fig. 8). Through the refinement of the model GranuVision3 for the analysis of calcium sparks measured with Fluo4 in cardiomyocytes⁴⁴, we were able to detect 298 TP and 2 FP events in comparison to 200 prior to refinement. This makes IVEA an extremely versatile and robust detection framework suited for a broader range of experimental conditions.”

Finally, we have promoted the benefits of the refinement training method through transfer learning in the discussion page 16, line 525-529 as follows: “These scripts facilitate the training of a new model and the implementation of transfer learning by freezing the majority of neural network layers and maintaining the last two Dense layers of the eViT. This refinement of an existing eViT model reduces the time required for training and the labeling of data. Additionally, it can be trained on a central processing unit (CPU) rather than a computer graphics card.”.

Thank you once again for your constructive feedback, we believe that our manuscript is much clearer after implementing your suggestions.

Overall, I found that the article difficult to follow with a lot of jargon which obscures the understanding. Mixing three methods for detecting fairly different types of imaging is confusing. It will be much clearer to describe the first type of recordings (non-fixed FBE) first, then add the particularities of the other types of recordings. Finally, the figures do not correctly describe the expected types of events and their treatment by the algorithms.

We thank the Reviewer for providing this feedback. We agree that originally our manuscript presented the three different analysis paradigms in a way that was not easily understandable. This was aggravated by the fact that the supplementary Fig. 1 panels were misquoted (for which we would like to apologize). We have rewritten the manuscript following the Reviewer’s suggestion to increase clarity. We also streamlined the introduction taking away all technical terms and just presenting the three analysis modules. These are the explained in detail in the first paragraph (“Classification of exocytic events for automated analysis”) of the result. There we present the three different type of exocytosis measurements giving rise to the three paradigms: random burst event (formally “non-fixed FBEs”), stationary burst events (formally “fixed FBEs”) and “hotspot area”. We then briefly explain the different analysis paradigms by specifying which one should be used for which type of measurement. Indeed, each type of exocytosis measurement should be analyzed by its specific analysis paradigm that is based on either our eViT (for the random burst events), the LSTM (stationary burst events) or the iterative thresholding coupled with k-means clustering (for the hotspot areas). The eViT and the LSTM have their own pre-trained models. While the eViT has two pretrained models, the LSTM has only one pretrained model. We now move to the Methods section all the detailed technical explanation and mathematical annotation to streamline the manuscript and thereby significantly reducing the amount of jargon.

To further reduce jargon, we now call these “random burst events”, “stationary burst events”, and “hotspot area”. The random burst events correspond to the exocytosis of individual fluorescent

vesicles anywhere on the plasma membrane. Usually these events are short lived and can be accompanied by a cloud of release fluorescent material. However the fluorescent signature of these type of events strongly depends on fluorescent protein used to label the vesicle. The stationary burst events correspond to the repetitive exocytosis of many fluorescent vesicles on the same location (such as pre-synapses in neurons). Their fluorescence signature correspond to a rise of fluorescence on more or less small spots (the synapse) for more or less short period of time depending of the length of the stimulus (see Supplementary Fig. 1). The “hotspot area” occur when measuring the spread of the released vesicular content using nanosensor-based technologies or nanofilms.

For example, Figure 1 shows “image patches”: are they extracts of the parent video, consecutive frames? What are the “image patches” leading to the event patterns depicted in the 13 regions? What is the rationale of choosing 13 regions?

Thank you for your insightful questions regarding Figure 1 and the rationale behind the 13 regions.

About the “image patches”: these are in principle simply cropped sections of video frames, created by isolating specific events rather than analyzing the entire video frame by frame. This approach captures localized activities over time within the regions of interest (ROIs). This is now explained in the manuscript page 18, lines 582-586 as: “Subsequently, an ROI is generated around each identified LM. These ROIs are then employed to generate image patches, which correspond to cropped sections of video frames. This approach captures localized activities over time, thereby enabling the isolation of specific events for classification rather than analyzing the entire video frame by frame by the neural network.”

About the 13 regions used in our LSTM analysis: these are a combination of one central region and four symmetrical regions, each further subdivided into three smaller subregions, which can be explained as follows:

Central Region:

Specifically designed to identify fusion events based on the mean intensity profile.

Symmetrical Regions:

Initially, circular regions were tested around the central region to study vesicle movement. However, this approach actually displayed reduced sensitivity to movement (see Supplementary Fig. 13a,c). To address this, the circular regions were split into four symmetrical regions, forming an orthonormal system. This division enhanced sensitivity to vesicle movement and increased the feature vectors fed into the LSTM network.

Segmentation:

Subdividing the symmetrical regions into three smaller subregions enabled the distinction between a fixed synapse and a moving vesicle, while capturing burst event activity (Supplementary Fig. 13a).

The selection of 13 regions originates from a practical balance between capturing significant spatiotemporal patterns and managing computational efficiency for the LSTM network. While this number is not absolute and alternative configurations may yield different results, this approach has consistently produced the most effective results for our analysis through the visualization of extracted features across various combinations.

This approach is now illustrated in the new Supplementary Figure 13 and mentioned in detail in the Methods section (page 20, lines 645-648) as follows: “The choice of small, symmetrical regions over circular masks enhances feature preservation of the spatiotemporal signal (Supplementary Fig. 13a,c). Additionally, we opted for 13 regions over 9 regions due to their higher sensitivity in capturing slow vesicle movements (Supplementary Fig. 13a,b)”.

For non-fixed FBE analysis, the authors present performance of their algorithm on 4 types of recordings with pH sensitive and pH insensitive probes (Figure 3). Presumably, the neural network models used for these analyses are different. This should be clearly explained.

The neural network that was used to analyze random burst events (formally non-fixed FBEs) is always the eViT with the same model namely the GranuVision2 (old model 1). This is now specified in Figure 3 column 4 of each panel on top of the predicted ROI image. However, for the new dataset with CD63-pHuji and CD9-SEP as marker, we used the GranuVision3 model as indicated in the Figure 3c. We have now amended the text to clearly mention this detail. We include this change in page 7 lines 246-249 in the manuscript as follows:

The model utilized was the GranuVision2, except for the CD63-phluorin, for which the GranuVision3 was used (Fig. 3). The GranuVision2 model was trained to differentiate between fusion with and without a cloud, while the GranuVision3 was trained on both phenomena, incorporating the latent granule fusion.

Moreover, the authors conclude that “IVEA detected 246 additional events from those identified by the HE [initially 673]”. The authors should describe if the events detected by IVEA are any different than the ones initially detected by human experts.

The events that were missed by the HE where either weak small events hard to see by eye, simply overlooked events due to possibly short attention span by the HE or events in which the cloud had a funny shape. This text has been incorporated into the manuscript, including the new dataset that was analyzed. This addition has increased the total number of events to 255, page 7, lines 238-242 as follows:

“Therefore, IVEA detected 255 additional events from those originally identified by the HE. All TP events were again manually verified by the HE to evaluate the network performance (Supplementary Table 2). The events that were originally missed by the HE where either weak small events hard to see by eye or simply overlooked possibly due to the limited attention span of the HE.”

Regarding style, half of the introduction, starting with “The event detection method...” describes the IVEA software. It should be in the Results section, with justification of the parameters used.

We agree with the reviewer. As indicated on the 2nd comment we completely rewrote the introduction thereby addressing this issue.

In conclusion, although a truly automated and efficient method is a major advance in the field of live cell fluorescence imaging, in particular for imaging exocytosis, the manuscript needs major rewriting before I can recommend it for publication.

We are extremely grateful for endorsing our work, we agree with all the points raised by the Reviewer and we have incorporated all the suggestions leading to a major improvement in the

description of the presented data, usability of the software, and importantly the readability of our manuscript.

Reviewer #1 (Remarks on code availability):

I could install the plugin, run it on my computer with data acquired in my lab. I could get some detected events but I have issues (see main review).

We are grateful for the time spent by the reviewer to set our software, we are happy to learn that the setup process was successful.

Reviewer #2 (Remarks to the Author):

Using machine learning, the authors developed a new method called “Intelligent Vesicle Exocytosis Analysis (IVEA)” to detect and analyze vesicle exocytosis events in living cells. IVEA integrates into the Fiji platform for open-source accessibility. The authors demonstrated superior capabilities of IVEA to detect vesicle exocytosis faster and more accurately in several images in different types of cells. Thus, IVEA will significantly help with imaging analyses of vesicle exocytosis, which will contribute to the scientific community. This manuscript could be worth publishing in Nature Communications. However, several issues should be addressed before it is accepted.

We are extremely grateful to our Reviewer for highlighting the potential value our work and we would like to thank them for the time spent reviewing our manuscript. We would like to mention that we have addressed all the raised issues in detail below.

One of major issues is whether IVEA can be applied to fluorescence images other than a pH-sensitive GFP to detect vesicle exocytosis events. As the authors may know, there are several other exocytosis indicators including FM, quantum dots and CypHer5E (please see the following references: (Murthy 1999, Hua, Sinha et al. 2011, Yu, Zhang et al. 2016, Jang, Kim et al. 2021, Park, Jung et al. 2022)). The authors should test whether IVEA can be used to detect vesicle exocytosis in images based on other exocytosis markers in order to support the authors’ claim – “IVEA has the potential to revolutionize the analysis of fluorescence fluctuations that occur during exocytosis, regardless of microscope type or fluorophore dye characteristics, with unprecedented accuracy and efficiency.”

This work presents a high performance analysis workflow “IVEA” able to detect exocytosis events, providing a wide number of exocytosis indicators and descriptors that can be used for further analysis, showcasing its versatility. IVEA was tested on videos in which exocytosing vesicles were labeled with genetically encoded markers that were pH sensitive (SEP and pHuji), weakly sensitive (mGFP and NeonGreen) and insensitive (tdTomato, mCherry) and via LysoTracker red (Supp. Fig. 4), which is a pH insensitive dye. As the Reviewer can understand, it is beyond the scope of this work to test IVEA with all the existing type of exocytosis indicators. Furthermore, since we offer a training platform that enable refinement of pre-trained models or the generation of new models, and we showed by the analysis of calcium sparks in cardiomyocytes (Supplementary Fig. 8) that the field of application of IVEA can be extended to other exocytosis sensors or many other burst like activities in the future by the user (see the conclusion section). However, we do agree with the Reviewer about the quoted statement, which has been down tuned to “a wide range of microscope types and fluorescent dyes “ in the completely newly written abstract. We hope that this satisfies the Reviewer.

Another issue is that the authors used similar imaging conditions for vesicle exocytosis for IVEA. Many laboratories used 1Hz imaging frequency to image vesicle exocytosis in living cells instead of 10Hz imaging. It is possible that IVEA may not properly detect vesicle exocytosis at different experimental conditions. Thus, the authors should test whether IVEA is capable of detecting vesicle exocytosis in images obtained at different experimental conditions.

We appreciate the reviewer’s concern regarding the ability of IVEA to detect vesicle exocytosis under different imaging conditions. To address this, we tested IVEA on datasets with reduced acquisition frequencies by down sampling 10 Hz videos to 5 Hz, 2 Hz, and 1 Hz using ImageJ’s stack reduction function. Our analysis confirmed that IVEA successfully detected exocytosis events even at 1 Hz without issue. However, as expected, decreasing imaging frequency resulted

in the natural loss of fast events that could no longer be observed, even by manual inspection. These events were effectively eliminated due to the lower temporal resolution, and as a result, IVEA did not detect them either. This confirms that IVEA's performance aligns with the fundamental limitations of temporal sampling rather than a failure of the algorithm itself. The new **Supplementary Figure 7** presents the results of this analysis, demonstrating IVEA's robustness across different acquisition frequencies. It is described in the manuscript page 9, lines 303-307 as follows: "In addition, we down-sampled 10 Hz videos down to 1 Hz to test IVEA performance at low image acquisition frequency. IVEA remains capable of detecting exocytosis at lower frequencies, but as the acquisition rate decreases, fast fusion events are naturally eliminated due to reduced temporal resolution, making them undetectable both manually and computationally (Supplementary Fig. 7 a–c)".

The other issue is that IVEA may not detect non-burst exocytosis events shown in hippocampal neurons. Not detecting non-burst exocytosis events in neurons could restrict the application of IVEA in neuroscience. I strongly encourage that the authors should further improve IVEA to detect non-burst exocytosis events in neurons for neuroscientists.

We appreciate the reviewer's suggestion regarding the detection of non-burst exocytosis events in neurons. Recognizing that this task requires different approaches and methods than those used for burst detection, we have updated the entire stationary burst event analysis module in IVEA and introduced the Vision Transformer network as an experimental alternative to the LSTM. The implementation of eViT required a new data handling strategy where we load data in blocks rather than all at once to optimize memory usage. We also developed a moving maximum intensity projection ("moving MIP") technique over a specified frame window to highlight gradual changes in synaptic activity, thereby improving the detection of slow exocytosis events. Once detected, the frame position of each event is aligned with the original video to ensure accurate mapping. This updated module is now available as a beta tool in the latest IVEA release, alongside the LSTM-based approach. We invite users to experiment with this feature as we continue to refine and extensively test it to improve the detection and classification of non-burst exocytosis events in neuronal imaging. This new method is not included in the manuscript due to its experimental and early-stage nature.

Here are some minor issues.

1. What stimulation protocols were used to cause fluorescent burst events (FBE)?

The stimulus protocol is different for each cell type and is given in the Methods section in the "Acquisition conditions" paragraphs with each cell type. In summary the murine CTL were stimulated by the anti-CD3 ϵ antibody coated coverslip, murine DRG neuron were stimulated via field electrode at 10 Hz for 30 s. Secretion in bovine chromaffin cells was induced through either depolarization trains or perfusion of the cells with a 5 μ M Ca²⁺ containing solution via the patch-clamp pipette, while in INS1 cells exocytosis was induced by superfusing the cells with a 75 mM K⁺ containing solution. For clarity, the information about the protocol has been added to the legend of figure 3 and 4b as follows: "Details on the exocytosis stimulation protocol are given in the Methods."

2. The text from line 72 to 166 should be placed in Results instead of Introduction.

We agree with the Reviewer and moved a large part of this section to the result as streamlined version (page 3, line 101-126). The part originally starting line 90 to 106 has been substantially shortened in the introduction removing most technical terms (now page 3, line 85-93). Thereby, we believe that these changes increased clarity and enhanced understanding.

3. The authors should describe how the authors calculate recall, precision and F1 score in details.

We thank the Reviewer for raising this point, we now describe this in detail in the Methods under the subsection named “Neural network training and evaluation” at pages 23-24, lines 779-800. For your convenience, we have added the answer on the model evaluation here. If you require a more comprehensive explanation of how IVEA exports labeled training data, please refer to our response to Reviewer 3’s question below, “How was the training of this network done?”

The added text in the manuscript is: “The final models were evaluated on videos unseen by the neural network, rather than reserving part of the training data for validation. A diverse array of datasets was utilized in the evaluation process, acquired from multiple laboratories employing a variety of microscopes. These included lytic granule exocytosis in T cells, dense core vesicle in INS-1 and chromaffin cells (both pH-sensitive and pH-insensitive fluorescent markers), dorsal root ganglion (DRG) neurons expressing Synaptophysin-SEP, and dopaminergic neurons examined with dopamine nanosensors (“AndromeDA”). The analysis showed strong concordance with human expert (HE) annotations. Most of the datasets were processed using default IVEA settings and automated parameter estimation, though users may override these defaults when necessary. For parameter estimation, these data sets are devoid of fusion events that occurred within the initial four frames of the videos, enabling IVEA to learn from. However, users have the option to disable automated learning and opt for manual override by adjusting the sensitivity to 1 or lower, thereby generating additional local maxima coordinates.

For each unseen video analyzed by IVEA, the resulting event ROIs were examined for validation. HE is labeling every detected as true exocytosis event (true positive, TP) or falsely predicted (false positive, FP). Any missed exocytosis event that was previously observed by the HE was labeled as a false negative (FN). Finally, precision, recall, and F1 score were computed based on the TP, FP, and FN counts, with the corresponding formulas:

$$Precision = \frac{TP}{TP+FP} \quad (12)$$

$$Recall = \frac{TP}{TP+FN} \quad (13)$$

$$F1\ Score = 2 \times \frac{Precision \times Recall}{Precision + Recall} \quad (14)$$

4. What is the size of the pixel in line 190?

The videos referenced in this section were simulated and created in MATLAB. Therefore, each pixel is treated as a unitless point. To make the simulated as close as possible to experimentally measured videos the size of the vesicle was comparable to the size in TIRFM acquired movies. This is now stated in the result section page 5, lines 165-168 as follows: “The initial evaluation of the software’s performance was conducted by the creation of simulated videos (Fig. 2b), employing a ground truth of vesicles with an average radius of 2.7 ± 0.36 non-dimensional pixels.

With this approach we emulated random burst events, as observed during the fusion of lytic granules in cytotoxic T lymphocytes (CTL) (see Methods)." Furthermore, we implemented this information in mathematical detail in the method section as well, page 25, lines 834-839 in the section "Video simulation and noise control".

To ensure clarity for all readers, we now include the pixel size information for the analyzed dataset and the relation with the neural vision radius. This information is added in page 7, lines 250-253. as follows: "The vision radius surrounding an event was adjusted according to the vesicle and its fusion size (see **Supplementary Fig. 5**). For instance, a radius between 14 and 16 pixels was used with videos that had a pixel size of 110 nm (**Fig. 3a-c**), while a radius between 7 and 12 nm was used for videos with a pixel size of 130 or 160 nm (**Fig 3d, e**)."

5. *Figure 2b, 4a and 5c were not mentioned in the main text.*

We apologize for this, we thank the Reviewer for spotting, we now cite these panels in the text at page 5, line 166 for 2b, page 10 line 336 for 4a, and page 13 line 448 for 5c.

6. *Font of "a, b, c, d" in main figures are too small.*

The letters of figure's panels have been corrected and are now labeled with 8-pt bold font instead of 7.

7. *Figure 5 used "(a), (b), ..." instead of "a, b, ..." in other figures. The authors should show the coherence in figures.*

We have now revised all the figures and figure captions and have unified the writing style.

8. *In line 519, "is capable to" → "is capable of"*

We thank the Reviewer for the correction. This is now implemented in the text.

References

- Hua, Y., R. Sinha, C. S. Thiel, R. Schmidt, J. Hüve, H. Martens, S. W. Hell, A. Egnér and J. Klingauf (2011). "A readily retrievable pool of synaptic vesicles." *Nat Neurosci* 14(7): 833-839.
- Jang, Y., S. R. Kim and S. H. Lee (2021). "Methods of measuring presynaptic function with fluorescence probes." *Appl Microsc* 51(1): 2.
- Murthy, V. N. (1999). "Optical detection of synaptic vesicle exocytosis and endocytosis." *Curr Opin Neurobiol* 9(3): 314-320.
- Park, C., S. Jung and H. Park (2022). "Single vesicle tracking for studying synaptic vesicle dynamics in small central synapses." *Curr Opin Neurobiol* 76: 102596.
- Yu, C., M. Zhang, X. Qin, X. Yang and H. Park (2016). "Real-time imaging of single synaptic vesicles in live neurons." *Frontiers in Biology* 11(2): 109-118.

We thank the Reviewer for taking the extra time to add the references. We added Ref 1, 2 and 3 to our introduction.

*Reviewer #2 (Remarks on code availability):
We tested IVEA and it works very well.*

We thank the Reviewer and their team for taking the time for testing and providing constructive feedback to us, leading to a major improvement to our software.

Reviewer #3 (Remarks to the Author):

With pleasure I was reading a manuscript about the developed "Intelligent Vesicle Exocytosis Analysis" (IVEA) platform for analysing vesicle fusion events in fluorescence microscopy images. Obviously, automation for these kinds of tasks is very useful, not just because they spare time, but also because it eliminates inter-observer variability. The fact that the authors developed their tool as Fiji plugin is exciting to note and will ensure that the community can adopt the tool.

We would like to thank the Reviewer for the kind and positive feedback and for the time they have spent reviewing our manuscript. We have now incorporated all the comments, leading to significant improvements in the quality of our work.

General feedback:

** As the tool was developed open-source for the Fiji platform, the authors could highlight early in the manuscript that it is available open source as common in the Fiji community.*

We agree with the Reviewer. The Abstract has been completely rewritten, and ImageJ is now mentioned as suggested page 1 Line 28 as follows: "We present the Intelligent Vesicle Exocytosis Analysis (IVEA) platform, an ImageJ plugin for automated, reliable analysis of fluorescence-labeled vesicle fusion events and other burst-like activities."

** The manuscript introduces multiple deep-learning based techniques, special new developed architectures and models. Hardly anything is written about how these were trained (e.g. line 242) and validated and which data was used for this. How many human experts were involved in producing the training data? What was their qualification to produce high-quality training data? Will the training data be available open access so that others can improve the state-of-the-art in the coming years?*

We apologize for not making this clear. The relevant experts who established the ground truth are now listed in Supplementary Table 2, which is cited in the manuscript page 22, line 737. They are the persons that performed the experiments to generate the analyzed video (15 people, mostly authors of this manuscript) with their respective supervisors (5 people, all authors of this manuscript), to ensure their familiarity with both the data and the assigned task. The verification of the results from IVEA was largely performed by Abed Chouaib assisted by Ute Becherer. In case of doubts, they referred to the person that acquired the data.

We trained the eViT network on 608 videos with 7,931 samples up to 24,916 with augmentation with 26x32x32 dimensions each. The LSTM was trained on ~90 videos, with a total of 11,300 samples with 13 x 41 dimensions sample each. The number of events on which the models were trained are indicted in the manuscript page 23 line 769-775 in the methods in the section *"Neural network training and evaluation"*. For more details, please refer to the Supplementary Table 7. As for the labeled data, we will upload the dataset with its corresponding labels information on open-source websites like datadryad.org, zenodo or similar websites that can store huge data up to 140 GB.

** Ablation studies: In computer science the concept of ablation studies is established to demonstrate that new methods are not overengineered or overfitting. If one develops a new algorithm / workflow that consists of many layers, paradigms and modules, single individual modules are deactivated in a test-run of the algorithm to see this step contributes to the overall performance of the tools. The presented tools are quite complex and I'm wondering if a combination of preprocessing + ViT + LSFTM + machine learning is really necessary to solve the scientific question. An ablation study might help.*

We appreciate the reviewer's suggestion regarding ablation studies to assess the necessity of each module in our pipeline. We'd like to specify that IVEA is never using the eViT together with the LSTM, and the k-means clustering for detection. These are three independent tools (i.e. modules) to analyze different type of events. For instance, non-fixed FBE (now called random burst events) were only classified with the encoder vision transformer (eViT) as it was superior compared to the LSTM. In contrast, the LSTM was used only to classify fixed FBE known for their long event duration (now called stationary burst events) and finally the k-means clustering with iterative thresholding module is applied only to events that correspond to hotspot area rather than burst events. The use of different modules was due to different constraints about specific features (random burst events, hot spot area) and high memory usage (stationary burst events). This is now clarified in the introduction page 3, line 85-93, in the result pages 3-4 line 120-126 and each subtitles displays the module that is used for the analysis.

We agree that an ablation analysis is common practice to investigate the performance of an algorithm. We conducted this analysis on the random burst events module since it contains many layers with a transformer block, which we systematically disabled to evaluate their impact on performance. As shown in **Supplementary Figure 15**, we performed the ablation study on the layers we have added to the ViT original architecture. We found that progressively removing these layers contributed in the overall network performance. We described it in details in the method section page 21, line 707-715 as follows: “The eViT architecture was subjected to ablation to study the impact of some layers on the model learning capabilities. This study was conducted by probing the model on an evaluation dataset, by eliminating layers and training the model again. The evaluation dataset was divided into two categories: exocytosis (positive labels) or non-exocytosis (negative labels). We performed the ablation study on the shared CNN layers and the penultimate Dense layer. The results show that progressively removing these layers leads to a noticeable decline in performance. The final configuration where only one convolution layer remains exhibited the strongest drop in performance. This study underscores the additive role of each layer in our current eViT architecture (see **Supplementary Fig. 15**).”

We further mention it in the discussion page 15, lines 505-508 as follows: “This necessitates the use of more sophisticated models, such as the ViT network. To support this model, an encoder network was adapted and implemented to extract features prior to the ViT network. The added layers to the ViT model³⁶ was demonstrated using an ablation study (Supplementary Fig. 15).”.

More detailed feedback:

Abstract:

** Line 23 "human capital" - maybe "human resources" or "work effort" might sound better?*

Thank you. This stamen has been amended to “significant human effort” in the completely rewritten abstract.

Introduction

* Line 43/44: "ion flux" depending on the target audience of the article, a sentence explaining what this is might be helpful. I'm a computer scientist and only have a vague idea.

We have changed the wording to "variation of ion concentration" that is much more accurate and understandable to a large public.

* Line 104: "which involves machine learning". After the deep-learning approaches in the lines before were more precise (using ViT and LSTM), I think it would be good to write here which algorithm was used as "machine learning".

We now add the following explanation at page 3, lines 91-93:

"Module three is designed to extract areas with fluorescent intensity variation using k-means clustering, and iterative thresholding."

* Line 105/106: Apologies for my ignorance, but I think an explanation of "nanosensor-based technologies" would be great.

We apologize for not explaining this more carefully, we have now added the necessary explanations in page 3, lines 115-120 as follows:

"Finally, a third category of events arise from the measurement of secreted transmitters using for example near infrared fluorescent dopamine nanosensor 'paint' AndromeDA^{35, 36}, and other composite nanofilms^{37, 38}. When the released neurotransmitter binds to the nanosensors around the synapse they emit a spreading fluorescent signal in a concentration-dependent manner (Supplementary Fig. 1f-g). We call these events "hotspot area" events."

* Line 108: "forward and backward subtraction" Could you explain what this is? Maybe an equation or a figure helps explaining the process. I suspect this is about image subtraction over time, but I'm not sure.

This is now explained in more detail in the manuscript in page 18 lines 571-576 in the methods section as follows:

"The event detection algorithm used by IVEA employs the grayscale image foreground detection method, which leverages a bidirectional image subtraction technique. This involves subtracting the intensity values in a 16-bit image stack of a reference frame I_i from an offset frame I_{i+n} in both directions—forward ($\Delta I_F = I_{i+n} - I_i$) and backward ($\Delta I_B = I_i - I_{i+n}$). This approach simplifies the handling of 32-bit matrices, reducing potential complexities with straightforward data processing."

* Line 115: "we select for each image regions": Is this a manual process? If not, what algorithm is used for this?

Thank you for pointing this out, the selection of image regions is fully automated. The algorithm detects local maxima (LM) as potential activity regions and extracts regions of interest (ROIs) around them. A global prominence, mean and FWHM threshold is applied, estimated from the first four frames and adjusted to prevent event loss. This threshold is intentionally lowered to balance between weak and high signals, and noise, while if excess events are detected, a neural network further refines the selection. We now clarified this, in page 18, line 581-589 as follows:

“To distinguish real events from noise, we detect local maxima (LM) in the subtracted images, representing potential exocytic hotspot. Subsequently, an ROI is generated around each identified LM. These ROIs are subsequently employed to generate image patches, which correspond to cropped sections of video frames. This approach captures localized activities over time, thereby enabling the isolation of specific events for classification rather than analyzing the entire video frame by frame by the neural network. The LM detection and ROI extraction process is fully automated, incorporating a global thresholding step that learns from the first four frames of the processed video. This ensures that noise and spurious maxima are filtered out, leaving only meaningful events for classification.”

** Line 118: "a ViT network, which classifies them as exocytosis or not". How was the training of this network done? How many images / annotations were necessary to train it?*

Thank you for your question and for pointing this out. We have expanded the manuscript pages 22-23 lines 736-766 to provide a detailed explanation of how the ViT network was trained, including the required number of images and annotations
Here are the main explanations.

“The initial data labelling and IVEA classification validation was performed by the human experts shown Supplementary Table 2.

These files include ROI center coordinates, frame positions, and radii. The IVEA software was used to export the labeled ROIs as zip files, with the training datasets being tagged as "_training_rois" to differentiate them from the evaluation data. The user can enable this option in the IVEA graphical user interface (GUI) in the “Custom Models” button. Prior to integrating the neural network with IVEA, the initial procedure involved exporting the selected regions identified by the automation processes and the manual labeling. Subsequent to the initial integration of the neural model, events in the training datasets were automatically labeled with a uniform naming convention that includes the list number, event ID, frame number, and classification category. For example, an event placed in the third position in the ROI manager list, with an ID of 3, detected at frame 779 and initially classified as category 1, would be named: "3-event {3} | frame 779_class_1." To prepare the data for the neural network training, we have developed a Python-based script with a simple interface and a JSON configuration file to extract or load the training data. The process of extracting the training data entails the following two steps. First, the ImageJ ROIs are read to identify the events positions and their categories. Subsequently, the times series patches are extracted at the position of each ROI. These data are then stored as Hierarchical Data Format (.h5) file format, organized into dictionaries containing "x_train" and "y_train" data, facilitating efficient loading and archiving of the training data.

During the training process, labels were refined through iterative process. Initially, the network was trained to distinguish between "exocytosis" and "not exocytosis." Later on, additional classes were introduced to differentiate between different exocytosis subtypes, as well as motion or noise artifacts. Real exocytosis classes received positive integers (e.g., 0, 1, 2), while non-exocytosis classes (such as noise or artifacts) were assigned negative integers (e.g., -1, -2). Whenever a misclassification was noted (for instance, class_1 instead of class_2 or -6), the label was corrected or a new one was defined, and the data were fed back into the network for retraining. To accommodate the substantial volume of events predicted and labeled by IVEA, a significant number of labels associated with non-exocytosis events were excluded to enhance data management. For generating a new model, training files and tool are available on our GitHub page for IVEA (see data available section).”

* Line 119: "developed for our ViT architecture (21)". The citation here suggests that the cited work was done by the same authors as the manuscript I'm reading. But I see no overlap between the author lists. May rephrasing this can clarify.

Thank you for bringing this to our attention. We acknowledge that the previous wording may have caused confusion regarding the authorship of the ViT architecture.

Here is the corrected text added at page 18-19, lines 604-606: "Each sequence is fed to a shared encoder layer attached to the ViT architecture for image recognition as described in Dosovitskiy et al.²¹. We designated the modified architecture as eViT."

Results

* Lines 168-180: This section reads like a Materials&Methods section and not Results.

We agree and removed most of it, kept the first sentence in the result section and moved the part on the parameter adjustment and PC requirement to the methods in section "Neural network training and evaluation" page 22, starting line 7312.

* Line 192: Same here. I would expect equations and explanations about what was done in a Methods section.

The Result section has been stripped of this information. They can now be found in the Methods section of the manuscript which has been updated to include further details regarding the incorporation of noise in the generated simulated videos. This addition was made under the 'Video simulation and noise control' topic page 25, lines 850-861. The following text has been included:

"For the noise control analysis, twenty videos with distinct SNRs were generated using MATLAB. Initially, a baseline video with no noise was created. Artificial white noise was then added to this baseline using `imnoise()` built-in function. Subsequently, the MATLAB built-in `poissrnd()` function was utilized to generate random photon shot noise commonly observed in microscopy. A Gaussian blur was applied to it to replicate the point spread function seen in microscopy images. Finally, the processed noise was added to the video to achieve the desired noise characteristics. The Poisson noise was modeled using the following equation:

$$I_{noise}(x, y) := Poisson(\lambda \cdot I(x, y)) \quad (17)$$

Where λ is the scaling factor that controls the relative level of noise.

To explore the impact of noise across a range of conditions, the scaling factor λ was varied incrementally from 0.1 up to 10 times the signal. Higher values of λ correspond to higher noise levels (lower SNR), while lower values of λ reduce the noise relative to the signal (higher SNR)."

* Line 258: The explanation of "event's cloud spread" also seems a method, not a result.

Thank you for your suggestion, now we have added a Gaussian non-max suppression topic in the methods section at page 22, lines 716-729, the text is added below:

"Gaussian non-max suppression"

Various non-maximum suppression techniques are typically used to address the multiple detection overlapping problem, including the classical intersection over union (IoU) method,

weighted boxes fusion (WBF), and others³⁰. However, these methods cannot be implemented with our data because exocytotic events cannot be limited to objects with boundaries or boxes. Thus, we have developed a new method that implements 3D Gaussian spread over time

$$g(x, y, t) = \Delta\mu \cdot e^{-\frac{(\Delta x^2 + \Delta y^2)}{2(\sigma + SR)^2}} \cdot e^{-\left(\frac{\Delta t^2}{2\tau^2}\right)} \text{ with } \tau = \nu \cdot \sigma \quad (10)$$

where $\Delta\mu$ is the fluorescence intensity of the event's area over subtracted image, σ the event's cloud spread, $SR = 1$ the spread radius controlled by the user, and event temporal cloud spread τ and ν the image acquisition frequency set to 10 Hz.

To find prime events from the redundant ones, we apply these criteria using $g(x, y, t)$ on both events such as:

$$\begin{cases} E_i = E_j & \text{if } \Delta\mu_i < g_j(x_j, y_j, t_j) \\ E_i \neq E_j & \text{otherwise} \end{cases} \quad (11)$$

where E_i and E_j are two different true positive (TP) events.”

** Line 291-311: This section is hard to read and follow. Would it be possible to put this data in a table and underline important results? E.g. the best method investigating a certain question.*

In order to respond to the aforementioned request, we removed from the result section the recall and precision values which we moved to a table that was incorporated into the supplementary data that contained the pertinent information (see: Supplementary Table 3). To enhance comprehension, a star rating system was included in the supplementary data to facilitate the evaluation of which neural network or software is most effective under which condition.

** Line 318: The comparison of the two developed method seems nice. I'm just curious if a comparison with methods developed by others would make sense?*

Yes, indeed, this comparison would make sense and we have done it in our manuscript. In detail we compared our method with the existing software “ExoJ” and “pHusion” for the random burst events, while for the stationary burst events we compared the “LSTM to SynActJ” plugin (see Fig. 3, Supplementary Table 3–6, and Supplementary Fig. 11).

** Line 324ff. Here comes a comparison with ExoJ but the context to the methods before is a bit wage. A short clear statement of all compared methods and their advantages and disadvantages would be great.*

Thank you for pointing this out. The text is now added in page 10, lines 321-330, and here: “ExoJ demonstrated an excellent performance comparable to that of IVEA on a specific type of exocytosis (latent vesicle fusion), while failing to detect other types. In contrast, IVEA’s deep learning-based approach, powered by eViT, successfully demonstrated a capacity to detect exocytosis events across both pH-sensitive and pH-insensitive datasets. By learning complex spatiotemporal patterns via new training or by adapting pretrained models via a refinement process, eViT adapts to diverse fluorescence signals (Supplementary Fig. 8). Through the refinement of the model GranuVision3 on only 30 calcium sparks events measured with Fluo4 in

cardiomyocytes⁴⁴, we were able to detect 298 TP and 2 FP events in comparison to 200 prior to refinement. This makes IVEA an extremely versatile and robust detection framework suited for a broader range of experimental conditions.”.

** Line 343: I presume, the CSV file also contains the coordinates of the center of mass?*

Thank you for your observation. For the stationary burst event analysis results, both the CSV files and the ImageJ ROI files currently include the center of mass. However, for the random burst event (formally non-fixed FBEs) and the hotspot area analysis, only the ImageJ ROI files provide the position of the events. Based on your suggestion, we have now implemented an option to include the center of mass for these events in the CSV files as well in the coming updates.

Reviewer #3 (Remarks on code availability):

I did not have the time to test the tool, the website offers tutorials and example data, which is a very good sign.

We thank the reviewer for their positive impression over the software website, tutorials and exemplary data. We are very grateful for the comments on the manuscript. By implementing them, we believe that our manuscript has significantly improved.

Point by point response to the reviewers

Reviewer #1 (Remarks to the Author):

I have read the revised manuscript together with the answer of the authors. I acknowledge the effort made by the authors to improve readability and better use of the plugin. I have no further comment on the manuscript, except another round of editing for numerous typos.

However, I find that the output given by IVEA, merely a list of event names, is minimal. The authors should consider adding more features for easy analysis of the data, such as the ones provided by the alternate plugin for analysis of exocytosis events ExoJ (local fluorescent measures, basic features of each exocytosis event, etc...).

Answer: Thank you for your positive review. We have addressed your request for more data analysis by including the spatial FWHM (x,y at peak frame), fluorescence rise time (10 to 90%), decay time (slope of the mono exponential decay fit), and temporal FWHM. We include this information in the manuscript page 7 lines 275 – 278 as follows: The CSV file contains measurements of the fluorescent intensity of each event over a fixed time intervals, with their x and y center of mass coordinates, their full width at half maximum (FWHM), and the fluorescence intensity kinetics (rise time, decay and temporal FWHM).

Finally, we have corrected numerous typos and grammatical mistakes by using ChatGPT reasoning model O3. As can be seen from the text version with the highlighted track change, we did not alter any sentence content in this process.

Reviewer #2 (Remarks to the Author):

It is disappointing that the authors did not apply IVEA to other exocytosis indicators including FM, quantum dots and CypHer5E because these exocytosis markers are widely used in many labs. I thought that this could be done by receiving raw images from research groups which have used FM, quantum dots and CypHer5E for their exocytosis assays. If the authors do not want to test whether IVEA is capable to detect exocytosis events in images from these FM, quantum dots and CypHer5E, the authors should mention the usage of these indicators to detect exocytosis and should write that IVEA has not been tested for these exocytosis indicators in Discussion of the revised manuscript clearly. Furthermore, the authors also should discuss the possibility of the application of IVEA to these endocytosis markers. Then, the readers will have better understanding of the application of IVEA.

It is discouraging that the authors did not include the new method to detect non-burst exocytosis events in neurons. The authors should discuss it in the revised manuscript. The discussion does not need to be lengthy. It will help the readers a lot.

Answer: We would like to thank you for your work on reviewer 3 comment. We were disappointed that we could not offer a simple solution for analyzing events in the hippocampus. However, we believe that giving users access to a beta version of IVEA eViT, which attempts to recognize synaptic transmission, could help neuroscientists with their analyses. Thank you for allowing us to address this issue in the discussion, which we have done on page 11, lines 428–436 as follows: However, IVEA is not designed to detect non-burst events such as those recorded in hippocampal neurons labeled with Synaptobrevin-SEP, FM1-43 or CypHer5E. In the case of Synaptobrevin-SEP stimulated exocytosis at synapses results in long-lasting increased fluorescence spreading out of the synapse^{50, 51}, while with FM1-43⁵² and CypHer5E⁹, it produces only a very slow decay of fluorescence. In the future, adapting

the eViT capability and retraining the new neural network may enable this type of analysis. However, this will require advanced computational capabilities and different dataset labeling. In contrast, we foresee that IVEA would have minimal difficulty of detecting synaptic transmission measured via iGluSnFR⁵³.

Finally, we are convinced that IVEA will be proficient in detecting mEPSPs measured with SynaptopHluorin 2× (DOI: 10.3389/fncel.2020.564081), since no extrasynaptic spreading of the signal should occur. However, we could not find an example video anywhere on the Internet or attached to a publication as supplementary data. Therefore, we cannot judge how effective IVEA would be in this task. This is why we decided not to mention it in the discussion.